# LLM-First Search: Self-Guided Exploration of the Solution Space

## Abstract

Large Language Models (LLMs) have demonstrated remarkable improvements in reasoning and planning through increased test-time compute, often by framing problem-solving as a search process. While methods like Monte Carlo Tree Search (MCTS) have proven effective in some domains, their reliance on fixed exploration hyperparameters limits their adaptability across tasks of varying difficulty, rendering them impractical or expensive in certain settings. In this paper, we propose **LLM-First Search (LFS)**, a novel *LLM Self-Guided Search* method that removes the need for pre-defined search strategies by empowering the LLM to autonomously control the search process via self-guided exploration. Rather than relying on external heuristics or hardcoded policies, the LLM evaluates whether to pursue the current search path or explore alternative branches based on its internal scoring mechanisms. This enables more flexible and context-sensitive reasoning without requiring manual tuning or task-specific adaptation. We evaluate LFS on Countdown and Sudoku against three classic widely-used search algorithms, Tree-of-Thoughts' Breadth First Search (ToT-BFS), Best First Search (BestFS), and MCTS, each of which have been used to achieve SotA results on a range of challenging reasoning tasks. We found that LFS (1) performs better on more challenging tasks without additional tuning, (2) is more computationally efficient compared to the other methods, especially when powered by a stronger model, (3) scales better with stronger models, due to its LLM-First design, and (4) scales better with increased compute budget. Our code will become publicly available upon acceptance.

## 1 Introduction

The reasoning and planning capabilities of Large Language Models (LLMs) have advanced significantly through increased test-time compute, akin to human *System 2* thinking, slow and deliberate, versus fast, intuitive *System 1* thinking (Kahneman, 2011). Early prompting techniques such as Chain of Thought (CoT) (Wei et al., 2022) enabled basic System 2 reasoning, but recent work reframes reasoning as a *search problem* (Koh et al., 2024b; Ye et al., 2025), leveraging classic algorithms such as Beam Search (Lowerre, 1976), Depth- and Breadth-First Search (DFS, BFS) (Knuth, 1998; Moore, 1959), Best-First Search (Hart et al., 1968), and Monte Carlo Tree Search (MCTS) (Coulom, 2006; Kocsis & Szepesvári, 2006). MCTS augmented with LLMs has proven effective across domains (Toma et al., 2021; Koh et al., 2024a; Zhou et al., 2023b) and is widely adopted. These systems often integrate LLM world models, reward/value estimators, self-consistency, self-refinement, multi-agent debate, and memory modules to achieve state-of-the-art (SotA) results (Hao et al., 2023; Zhou et al., 2023a; Murthy et al., 2023; Yu et al., 2024; Li et al., 2024; Gao et al., 2024; Qi et al., 2024; Di Zhang et al., 2024). A key limitation of MCTS is its sensitivity to the exploration-exploitation trade-off controlled by the exploration constant $C$ (Coulom, 2006; Kocsis & Szepesvári, 2006). Although hyperparameter tuning (Bischl et al., 2023) can optimise performance for a specific task, a fixed $C$ cannot

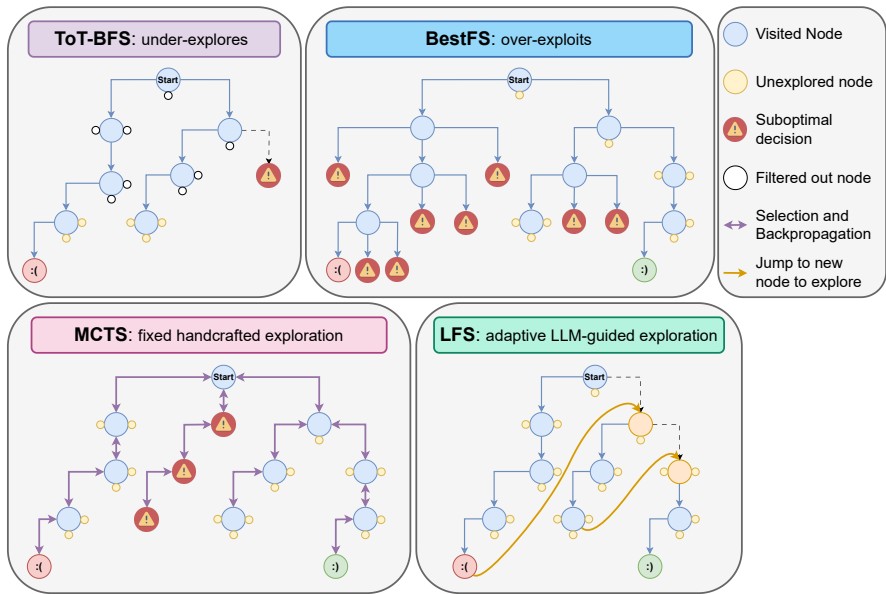

Figure 1: **Illustrative comparison of search strategies.** This figure visualises how different methods expand the search tree during reasoning. **Tree of Thought Breadth-First Search (TOT-BFS)** risks prematurely discarding promising paths due to rigid filtering criteria. **Best-First Search (BESTFS)** tends to over-exploit high-scoring nodes based on early estimations, potentially overlooking better long-term solutions. **Monte Carlo Tree Search (MCTS)** relies heavily on a fixed exploration constant, which can lead to either excessive exploration or over-commitment to suboptimal paths. In contrast, our proposed method, **LLM-First Search (LFS)**, removes the need for hand-tuned hyperparameters and handcrafted heuristics. Instead, it repurposes the LLM to both act and evaluate, enabling dynamic, model-guided decisions about whether to pursue the current reasoning path or explore alternatives. This tight integration between evaluation and exploration leads to more adaptive and efficient reasoning. A full search tree for both MCTS and LFS can be found in Appendix Section H. For clarity, the small circles (white and yellow) attached to the visited nodes refer the nodes' neighbours. Additionally, the dotted arrows refer to the edges that have not been traversed.

adapt to varying problem difficulties or LLM capabilities. Over-exploration hampers performance on simpler tasks where the LLM has strong priors, while under-exploration limits success on harder problems needing broader search (Dam et al., 2024; Sironi & Winands, 2021). This longstanding issue (Ruijl et al., 2013; Wang et al., 2020) parallels findings in Large Reasoning Models, which may overthink simple tasks due to excessive reliance on System 2 thinking (Ji et al., 2025; Zhang et al., 2025), analogous to MCTS's over-exploration from too high an exploration constant.

In this paper, we introduce **LLM-First Search (LFS)**, a novel approach that eliminates the need for manually tuned exploration hyperparameters, handcrafted heuristics, and traditional search algorithms. Building on recent MCTS extensions (Hao et al., 2023; Zhou et al., 2023a) and methods placing LLMs at the core of self-improvement (Zhang et al., 2023), LFS puts the LLM in control of the search process. Unlike MCTS, which relies on fixed exploration schedules, LFS lets the model autonomously decide whether to continue along the current path or explore alternatives based on its own evaluation, enabling adaptive, integrated exploration. A high-level depiction of how LFS works and how it overcomes the shortfalls of MCTS, as well as other well-established search algorithms, can be seen in Figure 1. We validate LFS on two reasoning

tasks, **Countdown** and **Sudoku**, showing competitive or superior performance with greater flexibility and adaptability than static search methods. Our main contributions are: (1) introducing **LLM-First Search**, a novel method that reimagines classical search by allowing the LLM itself to drive exploration, decision-making, and evaluation, removing the need for predefined search algorithms, (2) propose a **fully LLM-guided scoring and selection mechanism**, where the LLM evaluates whether the current search path is promising and dynamically decides to continue on this path or explore alternative paths, removing the need for manually tuned exploration hyperparameters, and (3) demonstrate, through experiments on **Countdown** and **Sudoku**, that LFS achieves competitive or superior performance relative to other popular search algorithms, while also demonstrating *greater efficiency, adaptability to task complexity, and scalability with increased model strength and compute budget*.

## 2    PRELIMINARIES

### 2.1    PROBLEM SETTING

**Markov Decision Process.** We consider problems that can be formulated as Markov Decision Processes (MDPs) (Bellman, 1957), where an agent interacts with an environment over a sequence of discrete time steps to achieve a goal. Formally, an MDP is defined by a tuple $(\mathcal{S}, \mathcal{A}, P, R, \gamma)$, where the agent observes a state $s \in \mathcal{S}$, selects an action $a \in \mathcal{A}$, transitions to a new state $s' \sim T(\cdot \mid s, a)$, and receives a reward $R(s, a)$.

**LLM Agents.** LLM agents are autonomous decision-making systems powered by large language models. Given an MDP, the LLM serves as a policy $\pi_\theta : \mathcal{S} \times \mathcal{T} \to \mathcal{A}$ parameterised by $\theta$, where $\pi_\theta(a_t \mid s_t, \mathcal{T})$ denotes the likelihood of taking action $a_t$ conditioned on the current state $s_t$ and task $\mathcal{T}$, to maximise the expected reward. These agents leverage language as a unified interface to perform environment understanding, reasoning and planning, and ultimately action execution (Wang et al., 2023; Mei et al., 2024; Huang et al., 2024). In our formulation, the LLM agent is provided with a natural language task description, the text description of the current state, and a list of valid next actions. The agent selects an action from this list, after which the environment deterministically transitions to a new state. This process is repeated until a terminal state is reached, at which point a reward is provided based on task success (e.g. win or lose). The specific MDP instantiations and prompts used for our two benchmark tasks, Countdown and Sudoku, are described in Section 5.2.

## 3    RELATED WORK

To enable models to reason more deeply and deliberately, researchers have developed a range of strategies, which we have broadly categorised as: (1) *Single-Shot Reasoning*, which elicits reasoning in a single prompt; (2) *Iterative and Reflective Reasoning*, which refines outputs through multiple steps; and (3) *Structured Search-Based Reasoning*, which treats reasoning as a search process. We briefly cover the first two, with a primary focus on the third, where our method lies.

**Single-Shot Reasoning.** Chain-of-Thought (CoT) prompting (Wei et al., 2022) encourages step-by-step reasoning via demonstrations, later simplified by minimal prompts like "think step by step," which elicit similar behaviour without examples (Zhang et al., 2022). Building on these foundations, several adaptations of these works have been explored (Kojima et al., 2022; Wang & Zhou, 2024; Xu et al., 2025). Recently, a "wait" token to slow down reasoning was introduced (Muennighoff et al., 2025), though it requires fine-tuning and is not purely an inference-time approach. Single-shot prompting has also been used to elicit more complex behaviours such as meta-in-context learning (Coda-Forno et al., 2023) and in-context distillation of algorithms like MCTS (Sel et al., 2023; Nie et al., 2024). While these methods have been effective on simpler tasks, they are inherently non-iterative and struggle to adapt to more complex tasks (Wang et al., 2022; Yao et al., 2023b; Madaan et al., 2023; Shinn et al., 2023).

**Reflective and Interactive Reasoning.** To go beyond linear reasoning, iterative and feedback-driven techniques have been proposed. A simple and widely used extension is self-consistency (Wang et al., 2022), which samples multiple CoT outputs and selects the most consistent answer. ReAct (Yao et al., 2023b) combines reasoning steps with task-specific actions and incorporates feedback to guide future steps. Other works refine LLM outputs through self-reflection or external feedback (Madaan et al., 2023; Shinn et al., 2023; Monea et al., 2024). Multi-agent debate frameworks (Du et al., 2023; Eo et al., 2025) further enhance reasoning by simulating dialogues between LLM agents to converge on a better final answer. However, these methods typically result in shallow exploration and lack explicit backtracking, limiting their ability to perform structured reasoning over long horizons or systematically explore multiple solution paths (Xie et al., 2023; Yao et al., 2023a; Koh et al., 2024b; Hao et al., 2023; Zhou et al., 2023a).

**Structured Search-Based Reasoning.** A growing line of work treats reasoning as a search problem, using classic search algorithms to guide LLMs through the task's search space, greatly improving the LLM's ability to solve complex reasoning and planning tasks. For example, Xie et al. (2023) proposes a stochastic beam search that samples and selects among multiple candidates at each step. Tree-of-Thoughts (ToT) (Yao et al., 2023a) introduces breadth-first and depth-first expansions of CoT-style reasoning, decoupling next-action selection and state value estimation. Several extensions have been proposed (Besta et al., 2024; Bi et al., 2024), though ToT remains the most prominent. Other works incorporate more advanced algorithms like Best-First Search (Koh et al., 2024b) and Monte Carlo Tree Search (MCTS) (Hao et al., 2023; Zhou et al., 2023a; Yu et al., 2024; Li et al., 2024; Gao et al., 2024; Qi et al., 2024; Di Zhang et al., 2024; Misaki et al., 2025). For example, RAP (Hao et al., 2023) uses MCTS with LLMs serving as a world model and a novel reward function composed of action likelihood and confidence, self-evaluation, and task-specific heuristics. LATS (Zhou et al., 2023a) extends RAP by incorporating environment feedback and reflective evaluation. More recent works integrate additional prompting strategies, such as reflection (Yu et al., 2024; Li et al., 2024; Gao et al., 2024) and multi-agent debate (Yu et al., 2024), for further performance gains. REX (Murthy et al., 2023) augments MCTS by allowing the LLM to perform multiple search steps, selection, expansion, and simulation, in a single response. The resulting actions are assigned rewards that are then backpropagated through each generated action. AB-MCTS (Misaki et al., 2025) introduces a novel node "GEN-node" which is a possible child for all nodes in the tree, which, if selected, prompts the LLM to create additional branches. While these methods have demonstrated strong performance, they are fundamentally built on traditional search algorithms that often rely on carefully tuned hyperparameters and handcrafted heuristics, limiting adaptability and requiring re-tuning for new tasks (Gao et al., 2024), rendering them impractical or very expensive for real use cases. Most recent works in this area represent incremental improvements to the base LLM-augmented variants of classic search algorithms, often incorporating additional prompting strategies like reflection or debate. Intelligent Go-Explore (IGE) (Lu et al., 2024) represents an important step in this direction, showing that LLMs can successfully drive exploration and achieving strong results on challenging benchmarks. However, IGE enforces fixed beam width and depth via static parameters, which limits adaptability across tasks. By contrast, our method removes such bounds entirely, allowing the LLM to decide dynamically when to backtrack or extend its search, thereby enabling truly self-guided exploration and reasoning. This shift addresses a core weakness of prior approaches while remaining compatible with, and likely to benefit from, incremental enhancements such as reflection or debate.

## 4 LLM-FIRST SEARCH (LFS)

In this section, we introduce **LLM-First Search (LFS)**, a method that empowers language models to *self-guide* their own search process by autonomously exploring and evaluating states and actions, enabling flexible, context-sensitive reasoning without manual tuning or task-specific adaptation. Specifically, given a task that can be initialised as a MDP, the LLM continuously interacts with the task environment, performing two key operations; (1) **Explore**, where it decides whether to continue along the current path or explore alternatives, and (2) **Evaluate**, where it estimates the value of each available action at the current state. We were able to

show that LLMs can effectively internalise and manage this process on their own, matching or exceeding the performance of traditional methods. The operations are detailed in the following paragraphs, with a high-level overview of LFS provided in Appendix Section B Algorithm 1.

**Exploration Decision.** At each step, given the current state $s_t$ and available actions $\mathcal{A}_t$, the agent is prompted with an exploration prompt $P_{\text{explore}}(s_t, \mathcal{A}_t)$ (the exact prompt can be seen in Appendix Section E) to decide whether to *exploit* the current path or to *explore* an alternative. If the agent chooses to *exploit*, it proceeds to the evaluation step using the actions in $\mathcal{A}_t$. Otherwise, if the agent opts to *explore*, it pops the highest-value node from the priority queue $\mathcal{Q}$:

$$(s'_t, \mathcal{A}'_t) \leftarrow \texttt{pop}(\mathcal{Q}),$$

and proceeds to the evaluation step using the new state $s'_t$ and corresponding actions $\mathcal{A}'_t$. This dynamic allows the agent to balance short-term commitment with broader exploration based entirely on its own internal judgment.

**Evaluate.** At each step, given a state $s_t \in \mathcal{S}$, a set of available actions $\mathcal{A}_t = \{a_t^1, \ldots, a_t^k\}$, and an evaluation prompt $P_{\text{eval}}(s_t, \mathcal{A}_t)$ (the exact prompt can be seen in Appendix Section E), the LLM is prompted to estimate the value $V(a_t^i \mid s_t)$ for each action, representing its utility or promise of leading to a high-reward solution. The best action is then selected:

$$a_t^* = \mathcal{A}_t \left[ \arg \max_i V_i \right]$$

$$\text{where } \{V_i\}_{i=1}^{|\mathcal{A}_t|} = P_{\text{eval}}(s_t, \mathcal{A}_t)$$

and executed, while all other candidate actions are added to a priority queue $\mathcal{Q}$ sorted by their estimated value. This structure enables efficient retrieval of high-potential alternatives in future exploration steps.

# 5 EXPERIMENTS

## 5.1 BASELINES

To ensure a fair comparison, all methods are evaluated using the same task setup and prompting format. We isolate the core effect of each search strategy by excluding incremental enhancements such as self-consistency, reflection, and debate, which are known to improve performance across many LLM-augmented approaches. Each method is tested with two models, GPT-4o and o3-mini (through the OpenAI API (OpenAI, 2024), with the configurations detailed in Appendix Section D), to assess performance across different model scales. We compare our approach against several strong LLM-augmented search baselines widely adopted in the literature. See Appendix Section B for baseline details.

**Thee-of-Thoughts Breadth-First Search (ToT-BFS).** Adapted from the setup in *Tree-of-Thoughts* (ToT) (Yao et al., 2023a), ToT-BFS expands a subset of child nodes up to a fixed depth. At each level, the LLM estimates the value of all child states, and only the top-$k$ states (with $k = 5$) are retained for further expansion. This process continues until a predefined maximum search depth is reached. Note that while ToT describe a DFS implementation, in our preliminary experiments, we found that DFS did not perform sufficiently (similar findings in (Yu et al., 2024)) and was therefore not considered further. In further support of this decision, in the ToT paper, they use countdown to test the BFS variant.

**Best-First Search (BestFS).** Following the approach in *Tree Search for Language model Agents* (Koh et al., 2024b), BestFS uses the LLM to estimate the value of the current state, which is then added to a priority queue. The next state to expand is selected greedily by popping the highest value from the queue. This process repeats until a solution is found or the search budget is exhausted.

**Monte Carlo Tree Search (MCTS).** Based on implementations from *RAP* (Hao et al., 2023) and *LATS* (Zhou et al., 2023a) and inspired by *AlphaGo* (Silver et al., 2016), we use PUCT to guide the MCTS algorithm.

Specifically, at each step, the LLM is used to (1) estimate a prior distribution over available actions at a given state, and (2) estimate the value of a leaf state after an action is simulated (the specific prompts used to elicit these behaviours can be found in Appendix Section E). These estimations are then integrated into the PUCT selection formula to balance exploration and exploitation. We performed a hyperparameter sweep over different exploration constants $C \in \{0.5, 1.0, 2.5\}$. The specifics of this can be found in Appendix Section F. We noted that $C = 0.5$ performed similarly to $C = 1.0$ in Countdown, but outperformed $C = 1.0$ in Sudoku (4x4), resulting in $C = 0.5$ achieving the best AUP.

## 5.2 TASKS

We evaluate our method and the baselines on two widely used reasoning and planning benchmarks: **Countdown** and **Sudoku**. They are widely adopted in the literature as reliable testbeds for evaluating structured reasoning with LLMs (Yao et al., 2023a; Zhou et al., 2023a; Ye et al., 2024; Seely et al., 2025). These benchmarks are particularly suitable for our evaluation for two key reasons: (1) **Scalability**, both Countdown and Sudoku allow for fine-grained control over difficulty, enabling evaluation across a spectrum of task complexities; and (2) **Complementarity:** Countdown offers a shallower search space with fewer steps, but selecting the correct action is often more challenging, even for humans. Conversely, Sudoku involves a much deeper search space with many more decision points, though it tends to be more intuitive for human solvers. Together, these benchmarks provide a balanced and comprehensive evaluation of search strategies across fundamentally different reasoning challenges. A more detailed discussion of the branching factors and widths of the two benchmarks can be found in Appendix Section C.

### 5.2.1 COUNTDOWN

Countdown (Wikipedia contributors, 2024) generalises the classic *Game of 24* (Yao et al., 2023a; Zhou et al., 2023a) and has become a challenging benchmark for evaluating LLM search due to its high branching factor and large combinatorial search space (Gandhi et al., 2024; Ye et al., 2024). The goal is to reach a target number $t$ using arithmetic operations $(+, -, \times, \div)$ applied to a list of numbers $n = [n_1, n_2, \ldots, n_l]$, where each number can be used at most once. For example, given $n = [1, 2, 3, 4, 5]$ and $t = 10$, a valid sequence is: $5 + 4 = 9, 3 - 2 = 1, 9 + 1 = 10, 1 \times 10 = 10$.

**Setup.** Following prior work (Yao et al., 2023a; Ye et al., 2024), we evaluate three difficulty levels with input lengths $l \in \{3, 5, 7\}$ and target $t$ sampled uniformly from $[10, 100]$. Each environment state $s_i$ is a 4-tuple $s_i = (t, n_i, o_i, A_i)$, where $t$ is the fixed target, $n_i$ is the current number set, $o_i$ the operation history, and $A_i$ the available actions. Each action $a \in A_i$ applies an arithmetic operation to two distinct numbers $n_j, n_k \in n_i$, producing a new number and modifying the set. The agent must find a sequence of actions that transforms $n$ into $t$. This setup naturally fits the MDP formalism: $\mathcal{S}$ is the space of number-operation configurations, $\mathcal{A}(s)$ the valid actions in state $s$, transitions modify the number set and operations based on the selected action, and the episode terminates on success or exhaustion of valid actions. The reward is 1 if the target is reached, and 0 otherwise. Prompting details are provided in Appendix Section E.

### 5.2.2 SUDOKU

Sudoku is a constraint satisfaction puzzle played on an $\ell \times w$ grid. The objective is to fill each cell with a value from a finite set $N = \{1, 2, \ldots, \ell \times w\}$ such that each value appears exactly once in every row, column, and subgrid. While the classic version uses a $9 \times 9$ grid with $3 \times 3$ subgrids, we generalise to arbitrary grid sizes, making Sudoku a rich, scalable benchmark for reasoning and search in structured environments.

**Setup.** We evaluate agents on two grid configurations: a $4 \times 4$ board (with $2 \times 2$ subgrids) and a more challenging $6 \times 6$ board (with $2 \times 3$ subgrids). Each environment state $s_i$ is defined as $s_i = (B_i, A_i)$, where $B_i \in \Sigma^{\ell \times w}$ is the current board and $A_i$ the set of valid actions. Each action $a \in A_i$ is a tuple $(x, y, v)$

assigning value $v \in N$ to cell $(x, y)$ without violating Sudoku constraints. Upon executing an action, the board is updated and valid actions recomputed. Episodes terminate when all cells are filled and constraints satisfied. As an MDP: $\mathcal{S}$ is the set of all valid partial boards, $\mathcal{A}(s)$ the set of valid $(x, y, v)$ assignments, transitions update the board, and reward is 1 if the final board satisfies all constraints, and 0 otherwise. See Appendix Section E for details on prompts used.

## 5.3 EVALUATION

### 5.3.1 METRICS

We evaluate each method over $n = 5$ runs per game at temperature $t = 0.0$, due to the stochasticity of LMs (Bender et al., 2021). Let $w_{i,j,r} \in \{0, 1\}$ indicate success of method $j$ on game $i$ in run $r$. The WinRate for game $i$ is

$$\text{WinRate}_{i,j} = \tfrac{1}{n} \sum_{r=1}^{n} w_{i,j,r},$$

with game$_{i,j}$ considered solved if $\text{WinRate}_{i,j} > 0.5$. Over all games $\mathcal{G}$, we report:

$$\text{WinRate}_j^* = \tfrac{1}{|\mathcal{G}|} \sum_{i \in \mathcal{G}} \text{WinRate}_{i,j}, \quad \text{EfficiencyScore}_j = \tfrac{\text{WinRate}_j^*}{\text{Tokens}_j^*},$$

where $\text{Tokens}_j^*$ is the average token usage of method $j$.

We compute 95% confidence intervals for $\text{WinRate}_j^*$ using the Wilson score interval (Wilson, 1927), preferred over the normal approximation for small $n$, which we report in the figures in Appendix Section G.

### 5.3.2 PERFORMANCE PROFILES AND AUP SCORE

Following Dolan & Moré (2002); Roberts et al. (2023); Nathani et al. (2025), we compare methods using performance profiles and their Area Under the Profile (AUP). For task set $T$ and method set $\mathcal{M}$, the performance ratio is $r_{t,m} = \max\{\ell_{t,m'} : m' \in \mathcal{M}\}/\ell_{t,m}$, where $\ell_{t,m}$ is the score of method $m$ on task $t$. The performance profile is

$$\rho_m(\tau) = \tfrac{1}{|T|} |\{t \in T : \log_{10}(r_{t,m}) \leq \tau\}|,$$

giving the fraction of tasks where $m$ is within $\tau$ (log-scaled) of the best method. The AUP is defined as $AUP_m = \int_1^{\tau_{\max}} \rho_m(\tau) \, d\tau$, where $\tau_{\max}$ is the smallest value for which all $\rho_m$ reach their maximum.

## 6 RESULTS AND ANALYSIS

### 6.1 TASK SPECIFIC

**Countdown.** In Table 1 we can see that in Countdown (Diff=3) all methods, except for TOT-BFS-GPT4O, are capable of solving 100% of the problems. TOT-BFS-GPT4O lags behind due to the lack of backtracking, compared to the other methods tested. Therefore, due to compute constraints, TOT-BFS-O3MINI is not tested. Additionally, no methods are tested with o3-mini in Countdown (Diff=3), as it is already near saturation with a weaker model. Following this, we can see that as we increase the difficulty of Countdown, TOT-BFS-GPT4O's WinRate drops drastically (72.64%) in comparison to BESTFS-GPT4O (50.53%), MCTS-GPT4O (40.0%), and LFS-GPT4O (36.84%). In Countdown (Diff=5) all backtracking methods are able to achieve a WinRate near or greater than 50%, with LFS-GPT4O marginally outperforming MCTS-GPT4O by 3.16%. LFS-GPT4O's improvement over the other methods increases even further in Countdown (Diff=7), **beating the next best method, MCTS-GPT4O, by a marked 14.74%, highlighting LFS's ability to scale better**

Table 1: WinRate (%) of each method across all tasks, evaluated with GPT-4o and o3-mini. LFS achieves the highest WinRates on all tasks for both models, except for Sudoku (4×4) when evaluated with GPT-4o.

| Model | Method | Countdown | | | Sudoku | |
| --- | --- | --- | --- | --- | --- | --- |
| | | Diff 3 | Diff 5 | Diff 7 | 4x4 | 6x6 |
| GPT-4o | ToT-BFS | 82.11 | 9.47 | 0.00 | 53.68 | 0.00 |
| | BestFS | **100** | 49.47 | 11.11 | 41.05 | 0.00 |
| | MCTS (c=0.5) | **100** | 60.00 | 32.63 | **100** | 0.00 |
| | MCTS (c=1.0) | **100** | 62.22 | 33.33 | 2.22 | 0.00 |
| | MCTS (c=2.5) | **100** | 60.00 | 24.44 | 0.00 | 0.00 |
| | LFS (OURS) | **100** | **63.16** | **47.37** | 96.84 | **2.22** |
| o3-mini | BestFS | – | 52.63 | 13.33 | 61.05 | 0.00 |
| | MCTS (c=0.5) | – | 69.47 | 41.05 | 90.53 | 4.21 |
| | LFS (OURS) | – | **70.53** | **78.95** | **96.84** | **25.26** |

Table 2: Area Under the Performance Profile (AUP), summarising the aggregate performance on all tasks. LFS achieves the best AUP score for all combination of metric and model.

| Metric | Model | ToT-BFS | BestFS | MCTS (c=0.5) | LFS (OURS) |
| --- | --- | --- | --- | --- | --- |
| WinRate | GPT-4o | 4.06 | 5.98 | 7.09 | **8.99** |
| | o3-mini | – | 4.23 | 6.00 | **7.20** |
| EfficiencyScore | GPT-4o | 3.68 | 2.67 | 3.68 | **4.70** |
| | o3-mini | – | 3.24 | 5.61 | **7.20** |

**as the task difficulty increases.** Note that all methods achieve a higher WinRate when using o3-mini in both Countdown (Diff=5) and Countdown (Diff=7), **with LFS-O3MINI again outperforming MCTS-O3MINI, especially in Countdown (Diff=7) by a significant 37.9%, indicating that LFS scales better with harder problems.** Interestingly, we can see that LFS's performance gain when using o3-mini is 39.17% (average % increase in WinRate over Countdown (Diff $\in \{5,7\}$), which is larger than the next best method, MCTS, which has a performance gain of 20.79%. **This shows that our method also scales better with stronger models.**

**Sudoku.** In Table 1 can see that in the simpler Sudoku (4x4), ToT-BFS-GPT4O again lags behind MCTS-GPT4O and LFS-GPT4O, however, outperforms BESTFS-GPT4O. This highlights one of the major drawbacks of BestFS, which is that it does not balance exploitation and exploration sufficiently, and in deeper and wider problems, where this becomes more important, BestFS falls behind. In Sudoku (6x6), all methods struggle to solve even a single game when using GPT-4o, with **LFS-GPT4O being the only method to achieve a WinRate greater that 0%, hinting as LFS's ability to scale with difficult tasks.** We can see that in Sudoku (4x4) BESTFS-O3MINI improves its WinRate (which makes sense since it is biased to over exploit, and is now guided by a stronger model), while LFS-O3MINI remains the same (likely due to it having been already close to saturation). Notably, MCTS-O3MINI's WinRate drops by 9.47%. This highlights a key limitation of MCTS: its performance is sensitive to the exploration constant $C$, which often requires retuning across tasks, difficulty levels, or base models, which is an expensive and impractical process. Lastly, we can see that **LFS-O3MINI's WinRate increases markedly in Sudoku (6x6), by 23.04%, beating the next best model, MCTS, by 21.05%, further highlighting LFS's ability to scale better with stronger models.**

## 6.2 KEY TAKEAWAYS

**Scalability and Improved Performance.** We highlight in the above that a key benefit of LFS is that it scales better as the difficulty of the problems increase, in contrast with BESTFS which does not balance exploitation and exploration adequately and MCTS which requires tuning for each task/model. Furthermore, LFS achieves a better WinRate, which again we highlight in the above discussion and can also be seen in Table 2 which shows that **LFS achieves the highest AUP values for WinRate, meaning that LFS has a higher performance on aggregate over all the tasks for both models**.

**Scaling with Stronger Models.** In the above analysis, we note that for Countdown (diff $\in \{5,7\}$), BESTFS, MCTS, and LFS see an improvement in their performance when using a stronger model. LFS, however, has a notably much larger performance increase when playing the most difficult version of Countdown. In fact, it performs even better in Countdown (diff=7) than Countdown (diff=5). Interestingly, we note that when using o3-mini, MCTS actually sees a decrease in performance in Sudoku (4x4) (we hypothesise that this is due to o3-mini overestimating state values, which leads to poorer exploration), compared to BESTFS' increase and LFS' stability. In Sudoku (6x6), LFS again has a notably larger performance increase compared to MCTS. All together, these results show that **LFS scales better with a stronger model, compared to the other methods**.

**Scaling with Increased Compute and Computational Efficiency.** We found that as the token usage increases, the total number of Countdown games won increases, with LFS distinctly outperforming the next best method, MCTS. This can be seen in Figure 21 in the Appendix. This trend is particularly notable for LFS with o3-mini since it scales better with a stronger model, and thus the gap between our method and the others, increases. Note that due to compute limitations, we could not test each method for larger token limits, but we can see that the gap between our method and the others is likely to continue to grow, if the current trend continues. We can see a similar trend for the Sudoku games won in Figure 21 in the Appendix, however less prominent due to the WinRate saturation for the simpler Sudoku version and the poorer performance for the harder Sudoku. Lastly, not only does our method scale better with compute, it is more computationally efficient. We can see this in Table 2, where LFS achieves the highest AUP score for EfficiencyScore, which as discussed in Section 5.3, represents the models' computational efficiency.

## 6.3 QUALITATIVE ANALYSIS

While the quantitative results already show that LFS is the strongest performer, a qualitative examination of the decision traces reveals why this advantage arises. The key distinction is that LFS replaces parametrised, heuristic-driven exploration with self-guided exploration. This shift allows LFS to navigate wide and deep combinatorial spaces where methods like MCTS, the next best performing framework, systematically fail.

### 6.3.1 THE PARAMETRIC TRAP

MCTS depends on the PUCT rule, where the exploration-exploitation balance is controlled entirely by a single scalar constant, $c$. The behaviour of MCTS, and its failure modes, are therefore tightly linked to this hyperparameter. In complex reasoning tasks, where simple heuristics do not predict solvability, this introduces structural weaknesses that LFS avoids through agentic, context-aware decision-making.

**Risk 1: Under-Exploitation** If $c$ is too low, MCTS behaves greedily. This is particularly troublesome in tasks where states appear promising under shallow heuristics (for example, numerically close to the target which is consistently used by the models) but are in fact logical dead ends. In our traces, MCTS repeatedly over-commits to such states, simulating unproductive subtrees because the heuristic dominates the PUCT score. LFS, by contrast, explicitly reasons about future reachability rather than surface-level heuristics. For example:

> "We are only 1 away from the target... However, we do not have a direct way to create the
> number 1... It seems more beneficial to explore a new state."

LFS chooses exploration not because a formula forces it, but because it has understood that the current path is impossible to complete.

**Risk 2: Over-Exploration**   If $c$ is too high, the opposite failure occurs: MCTS is forced to over-explore. Promising but paths, which require several steps that may temporarily worsen the heuristic signal, are abandoned early simply because the exploration term dominates. LFS, however, stays committed to such paths when they remain semantically valuable. For example:

> "Even though the current number is far from the target, creating small integers is valuable
> for fine-tuning in later stages."

Here, LFS persists because it has reasoned that the long-term plan retains value. MCTS, governed by visit counts and short-term rewards, cannot represent or preserve this form of long-horizon intent without task-specific engineering.

### 6.4 Convergence in Simple Domains

In simple tasks where heuristics align closely with solvability, the behavioural gap between LFS and MCTS diminishes. For instance, in the example with *Target: 80* and *Numbers: [7, 60, 27]*, the solution path is shallow and immediately rewarded. LFS identifies this at Step 0:

> "The most promising operation is '60 + 27 = 87', as it gets us very close to the target... The
> operation '87 - 7' directly reaches the target number."

MCTS performs similarly here because the heuristic is reliable and the reward signal is immediate. However, such cases are the exception. Once the problem becomes wide, deep, or misleading under simple heuristics, the divergence becomes clear: MCTS collapses into either greedy over-commitment or forced over-exploration, depending on the choice of $c$. LFS, by contrast, remains stable because exploration is guided by reasoning.

## 7   Conclusion

In this paper, we introduced **LLM-First Search (LFS)**, a novel approach to reasoning and planning that places the language model itself at the core of the search process. Unlike traditional search methods such as MCTS, BestFS, or BFS, which rely on external heuristics, fixed traversal strategies, or carefully tuned hyperparameters, LFS empowers the LLM to autonomously determine whether to continue down a path or explore elsewhere in the tree, using only its internal reasoning and planning capabilities, which we term *Self-Guided Search*. Through experiments on two complementary benchmarks, Countdown and Sudoku, we demonstrated that LFS offers several key advantages: (1) stronger performance on harder instances without task-specific tuning, (2) improved computational efficiency, particularly with more capable models, (3) better scalability with model strength, and (4) greater responsiveness to increased compute budget. These findings validate LFS as a flexible, LLM-centric framework that not only outperforms classic search methods but also adapts more naturally to varying task complexity and compute budgets. By unifying decision-making and evaluation within the LLM itself, LFS reimagines the role of search in LLM reasoning, not as a separate, manually controlled process, but as an integrated, language-driven mechanism. This shift enables a more general, adaptable, and efficient form of reasoning, offering a promising direction for scalable LLM-based problem solving. While our evaluation was limited to a subset of tasks and models due to compute constraints, it serves as a starting point for future work to extend *LLM-First Search* to more complex and realistic settings, where its benefits in adaptive exploration and *self-guided* reasoning are likely to be even more pronounced.

## 8 REPRODUCIBILITY

We have taken care to make our work reproducible. The main text and appendix provide full implementation details. Exact prompts and experimental configurations are included to enable replication of our results. An open-source codebase with detailed instructions will be released upon acceptance, ensuring that all experiments can be reproduced and extended by the community.

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

## APPENDIX TABLE OF CONTENTS

## A    LIMITATIONS AND FUTURE WORK

We evaluate our method, LLM-First-Search (LFS), on two standard reasoning benchmarks: **Countdown** and **Sudoku**, commonly used in Large Language Model (LLM) research. These tasks offer (1) **scalability**, allowing fine control over difficulty, and (2) **complementarity**, with Countdown featuring a shallow but challenging search space, and Sudoku a deeper but more intuitive one. Together, they provide a balanced testbed for search strategies across diverse reasoning challenges. However, these benchmarks lack some complexities of real-world problems. Due to compute constraints, we limited our experiments to these tasks and a fixed number of samples, restricting broader validation. LFS also assumes the ability to revert to previous states, which may not hold in all environments. Additionally, while LFS is shown to excel with stronger language models, we did not determine its sensitivity to weaker models. While our evaluation was limited, it serves as a starting point for future work to extend *LLM-First Search* to more complex and realistic settings, where its benefits in adaptive exploration and *self-guided* reasoning are likely to be even more pronounced.

# B   ADDITIONAL DETAILS OF SEARCH BASELINES

## B.1   LLM-FIRST SEARCH (LFS)

The LLM-First Search (LFS) is shown is summarised in Algorithm 1 below.

---

**Algorithm 1** LLM-First Search (LFS)

---

1: **Input:** LLM $\pi_\theta$, Prompts $P_{eval}$ and $P_{explore}$, Transition function $T$
2: Initialise $s_0$, $\mathcal{A}_0$, Priority queue $\mathcal{Q}$
3: $\{V_i\}_{i=1}^{|\mathcal{A}_0|} = P_{\text{eval}}(s_0, \mathcal{A}_0, \pi_\theta)$
4: $a_0^* = \mathcal{A}_0 [\arg\max_i V_0]$
5: $\mathcal{Q} := \mathcal{Q} \cup \{a \in \mathcal{A}_0 | a \neq a_0^*\}$
6: $(s_1, \mathcal{A}_1) \sim T(\cdot \mid s_0, a_0^*)$
7: $t = 1$
8: **while** Token limit not exhausted **do**
9:     **if** $P_{explore}(s_t, \mathcal{A}_t, \pi_\theta)$ **then**
10:         $(s_t', \mathcal{A}_t') \leftarrow \text{pop}(\mathcal{Q})$
11:     **else**
12:         $\{V_i\}_{i=1}^{|\mathcal{A}_t|} = P_{\text{eval}}(s_t, \mathcal{A}_t, \pi_\theta)$
13:         $a_t^* = \mathcal{A}_t [\arg\max_i V_i]$
14:         $\mathcal{Q} := \mathcal{Q} \cup \{a \in \mathcal{A}_t | a \neq a_t^*\}$
15:         $(s_t', \mathcal{A}_t') \sim T(\cdot \mid s_{t'}, a_{t'}^*)$
16:     **end if**
17:     $(s_t, \mathcal{A}_t) \leftarrow (s_t', \mathcal{A}_t')$
18:     $t \leftarrow t + 1$
19: **end while**
20: **Return:** $(s_t, \mathcal{A}_t)$

---

## B.2   TREE-OF-THOUGHT BREADTH-FIRST SEARCH (ToT-BFS)

In this section, we describe **Tree-of-Thought Breadth-First Search (ToT-BFS)**, a method inspired by the Tree-of-Thought framework. ToT-BFS performs uniform expansion from the current frontier: at each depth level, it evaluates all current frontier nodes and expands the top-$k$ according to their LLM-estimated value. The method is summarised in Algorithm 2.

**Frontier Filtering.**   At each iteration, the search maintains a set of current frontier nodes $\mathcal{F}_t = \{(s_t^1, \mathcal{A}_t^1), \ldots, (s_t^n, \mathcal{A}_t^n)\}$ representing all active paths at the current depth. For each node, the LLM is used to score the value of the state via a prompt $P_{\text{eval}}(s_t^i)$, returning an estimated utility $V(s_t^i)$. The top-$k$ nodes with the highest estimated value are selected for expansion:

$$\mathcal{F}_t^{top} = \text{TopK}(\mathcal{F}_t, \{V(s_t^i)\}),$$

where each selected node is expanded by executing actions from $\mathcal{A}_t^i$ using the environment's transition function $T$. If the frontier node with the highest estimated value is terminal, the expansion ends, and the terminal state is returned.

**Frontier Expansion.**   Each selected frontier node $(s_t^i, \mathcal{A}_t^i)$ is expanded, resulting in new states $(s_{t+1}, \mathcal{A}_{t+1})$ which are added to the new frontier. This process continues level by level, maintaining a breadth-first structure that allows the model to explore multiple solution pathways in parallel.

---

**Algorithm 2** Tree of Thought Breadth-First Search (ToT-BFS)

---

1: **Input:** LLM $\pi_\theta$, Value prompt $P_{\text{eval}}$, Transition function $T$, Beam width $k$
2: Initialise frontier $\mathcal{F} := \{(s_0, \mathcal{A}_0)\}$
3: **while** Token limit not exhausted **do**
4:   Evaluate all frontier states: $\{V_i = P_{\text{eval}}(s_i, \pi_\theta)\}_{i=1}^{|\mathcal{F}|}$
5:   Select top-$k$ states by value: $\mathcal{F}^{top} \subseteq \mathcal{F}$ with $|\mathcal{F}_{top}| = k$
6:   $(s_t, \mathcal{A}_t) \leftarrow \mathcal{F}^{top}\left[\arg\max_{(s, \mathcal{A}) \in \mathcal{F}^{top}} V(s)\right]$
7:   **if** $s_t$ is terminal **then**
8:     **break**
9:   **end if**
10:   Initialise new frontier $\mathcal{F}_{new} := \emptyset$
11:   **for** each $(s_i, \mathcal{A}_i) \in \mathcal{F}^{top}$ **do**
12:     **for** each $a \in \mathcal{A}_i$ **do**
13:       $(s', \mathcal{A}') \sim T(\cdot \mid s_i, a)$
14:       $\mathcal{F}_{new} := \mathcal{F}_{new} \cup \{(s', \mathcal{A}')\}$
15:     **end for**
16:   **end for**
17:   $\mathcal{F} := \mathcal{F}_{new}$
18: **end while**
19: **Return:** $(s_t, \mathcal{A}_t)$

---

## B.3 BEST-FIRST SEARCH (BESTFS)

---

**Algorithm 3** Best-First Search (BestFS)

---

1: **Input:** LLM $\pi_\theta$, Value prompt $P_{\text{eval}}$, Transition function $T$
2: Initialise $s_0$, $\mathcal{A}_0$, Priority queue $\mathcal{Q}$
3: Evaluate current state: $V_0 = P_{\text{value}}(s_0, \pi_\theta)$
4: $\mathcal{Q} := \mathcal{Q} \cup \{(V_0, s_0, \mathcal{A}_0)\}$
5: **while** Token limit not exhausted **do**
6:   $(V_t, s_t, \mathcal{A}_t) \leftarrow \text{pop}(\mathcal{Q})$                    ▷ Greedy selection by highest $V_t$
7:   **for** $a_t \in \mathcal{A}_t$ **do**
8:     $(s', \mathcal{A}') \sim T(\cdot \mid s_t, a_t)$
9:     $V' = P_{\text{value}}(s', \pi_\theta)$
10:     $\mathcal{Q} := \mathcal{Q} \cup \{(V', s', \mathcal{A}')\}$
11:   **end for**
12: **end while**
13: **Return:** $(s_t, \mathcal{A}_t)$

---

In this section, we describe **Best-First Search (BestFS)**, a strategy that expands the most promising nodes first, based on their estimated value. Our implementation leverages an LLM to evaluate the value of states and uses these estimates to drive the search greedily toward high-reward regions of the search space. BestFS does not prompt the LLM to decide when to explore; rather, it always expands the node with the highest estimated value from the priority queue. A high-level overview is provided in Algorithm 3.

**LLM-Based Evaluation.**   The LLM is prompted using a value-estimation prompt $P_{\text{eval}}(s')$, to evaluate the state $s'$ after taking action $a_t \in \mathcal{A}_t$ which returns a scalar estimate $V'$ of the utility of $s'$. The tuple $\{(V', s', \mathcal{A}')\}$ is then added to the priority queue $\mathcal{Q}$. This is done for all $a_t \in \mathcal{A}_t$.

**Greedy Expansion.**    At each step, the algorithm pops the highest-ranked node $(s_t, \mathcal{A}_t)$ from the priority queue $\mathcal{Q}$:

$$(s_t, \mathcal{A}_t) \leftarrow \texttt{pop}(\mathcal{Q}),$$

where $\mathcal{Q}$ is ordered by the estimated value of states as predicted by the LLM.

### B.4   Monte Carlo Tree Search (MCTS)

---

**Algorithm 4** LLM-guided Monte Carlo Tree Search (MCTS)

---

1: **Input:** LLM $\pi_\theta$, Prompts $P_{\text{prior}}$ and $P_{\text{value}}$, Transition function $T$
2: Initialise root node $s_0$
3: **while** Token limit not exhausted **do**
4:      $path \leftarrow []$
5:      $s \leftarrow s_0$
6:      **while** $s$ is not leaf and not terminal **do**
7:          $a \leftarrow \text{PUCT}(s)$                                      ▷ Uses visit counts and priors
8:          $path \leftarrow path \cup \{(s, a)\}$
9:          $s \leftarrow T(s, a)$
10:      **end while**
11:      **if** $s$ is leaf **then**
12:          $\mathcal{A} \leftarrow \text{actions}(s)$
13:          $\{P(a \mid s)\} \leftarrow P_{\text{prior}}(s, \mathcal{A}, \pi_\theta)$
14:          $V(s) \leftarrow P_{\text{value}}(s, \pi_\theta)$
15:          Initialise state statistics: $\{P(a)\}^{\mathcal{A}}, V(s), N(s)$
16:      **end if**
17:      **if** $\texttt{is\_solution}(s)$ **then**
18:          **break**
19:      **end if**
20:      Backpropagate $V(s)$ along $path$
21: **end while**
22: **Return:** $(s, \mathcal{A})$

---

In our adaptation of **Monte Carlo Tree Search (MCTS)**, we replace traditional simulation-based rollouts with value and policy estimates provided directly by the LLM. Specifically, at each node, the LLM is prompted to estimate (1) the value of the current state, and (2) the prior over the available actions, which are used by the **PUCT** selection rule to guide the search. The resulting algorithm is outlined in Algorithm 4.

**Search Tree and Node Structure.**    MCTS maintains a search tree where each node corresponds to a state $s$, and stores the visit count $N(s)$, total value $W(s)$, and prior over actions $\{P(a \mid s)\}$ (as returned by the LLM). Each edge stores a running estimate of $Q(s, a) = W(s, a)/N(s, a)$. The tree is expanded progressively, guided by the PUCT criterion:

$$a^* = \arg\max_a \left[ Q(s, a) + c_{\text{puct}} \cdot \pi(a \mid s) \cdot \frac{\sqrt{N(s)}}{1 + N(s, a)} \right],$$

where $c_{\text{puct}}$ is the exploration constant controlling the trade-off between exploration and exploitation. This selection rule encourages the algorithm to prioritise actions with either high expected value or low visitation count, as informed by the LLM's prior.

**LLM-Based Evaluation.** To avoid traditional rollout-based playouts, we leverage the LLM to provide value and policy estimates directly at the leaf node. When a new leaf node is reached, we prompt the LLM using a state-value prompt $P_{\text{value}}(s)$ to obtain a scalar estimate $V(s)$ of the state's expected utility. We also query an action-prior prompt $P_{\text{prior}}(s, \mathcal{A})$ to estimate the prior distribution over actions. These values are then backpropagated through the tree to update $Q$, $W$, and $N$ values for all nodes along the visited path.

## C  TASK DISCUSSION AND ANALYSIS

We analyse the branching factor and number of states at a given depth $d$ for our two benchmark tasks, Countdown and Sudoku, demonstrating their complementary characteristics. This analysis supports the use of these tasks as representative testbeds, with Countdown exhibiting a shallower but more complex decision space and Sudoku presenting a deeper, broader search space, together providing a balanced evaluation of search strategies.

**Countdown.** Starting with an initial list of $n$ numbers, at each step the agent selects two distinct numbers and applies one of four arithmetic operations $(+, -, \times, \div)$. The number of distinct pairs is $\binom{n}{2} = \frac{n(n-1)}{2}$, and each pair can be combined with 4 possible operations. Thus, the branching factor at the root (depth $d = 0$) is:

$$B_0 = 4 \times \binom{n}{2} = 2n(n-1).$$

After applying one operation, the list size decreases by 1, leaving $n - 1$ numbers. At depth $d$, the list size is $n - d$, so the branching factor at depth $d$ is:

$$B_d = 4 \times \binom{n-d}{2} = 2(n-d)(n-d-1).$$

The number of distinct lists (states) exactly at depth $d$, denoted $L_d$, can be recursively computed as:

$$L_0 = 1,$$

$$L_d = L_{d-1} \times B_{d-1} = \prod_{i=0}^{d-1} 2(n-i)(n-i-1).$$

**Sudoku.** In a Sudoku puzzle of size $l \times l$, assume $n$ empty cells initially. At each step, the agent fills one empty cell with a valid number (up to $l$ possibilities).

At depth $d$, there are $n - d$ empty cells left, so the branching factor is:

$$B_d = (n-d) \times l.$$

The number of board states exactly at depth $d$ is then:

$$L_0 = 1,$$

$$L_d = L_{d-1} \times B_{d-1} = \prod_{i=0}^{d-1} (n-i) \times l = l^d \times \prod_{i=0}^{d-1} (n-i).$$

**Analysis.** Countdown features a relatively shallow search space with a maximum depth of $n - 1$, where $n$ is the initial length of numbers in the set. At each depth $d$, the branching factor is given by

$$2(n - d)(n - d - 1),$$

reflecting the number of possible pairs and arithmetic operations. Although the search depth is limited, Countdown is often more challenging in terms of selecting the correct action due to the combinatorial nature of valid operations.

In contrast, Sudoku involves a much deeper search space, with maximum depth equal to the initial number of empty cells $n$. The branching factor at depth $d$ is approximately

$$(n - d) \times l,$$

where $l$ is the board's side length (e.g., 9 for a standard $9 \times 9$ Sudoku). Here, the width of the search space depends linearly on the number of remaining empty cells and the number of valid entries per cell, resulting in a wide and deep search tree.

This contrast in search space structure, Countdown's shallow but combinatorially complex branching versus Sudoku's deep and broadly branching tree, makes these benchmarks complementary, providing a thorough evaluation of search strategies under diverse reasoning challenges.

## D    IMPLEMENTATION DETAILS

We utilised the OpenAI API to access both the GPT-4o and o3-mini language models. We set key parameters while leaving others at their default values. The *temperature* was fixed at 0.0 to produce deterministic outputs and reduce randomness. We set *max_tokens* to 16,384 to allow sufficiently long responses for complex, multi-step reasoning tasks. A *timeout* of 300 seconds was enforced to limit API call duration and prevent excessively long requests. Lastly, the o3-mini model was configured to operate at a "low" *reasoning_effort*.

## E    PROMPTS

This section presents the exact prompts used in our experiments. These prompts were designed to guide the language model in performing evaluations, making exploration decisions, or generating actions during search. These prompts play a crucial role in enabling *LLM-First Search* and the other baselines to operate under comparable conditions, ensuring that differences in performance arise from the methods themselves rather than discrepancies in task formulation. Note that variables enclosed in curly braces (e.g., {state}, {actions}) indicate Python variables used for string formatting (this will be visible in the accompanying open-source code). Lastly, for clarity, we use colour to distinguish different components of the prompts: (1) Green: Task-specific instructions or rules, (2) Red: System-level instructions that define the model's role or behaviour, and (3) Blue: User-level queries or task inputs.

### E.1    COUNTDOWN

> *Countdown Game Rules*
>
> You're playing the Countdown Numbers Game. Let me explain the rules and how to solve it:
> **Game Rules:**
>
>   1. You are given a set of numbers and a target number to reach.
>   2. You can only use each number once.

3. You must combine numbers using only four operations: addition (+), subtraction (-), multiplication (*), and division (/).

4. Division is only allowed when it results in a whole number (no fractions or decimals).

5. You can only combine two numbers at a time to create a new number.

6. After each operation, the original numbers are removed, and the result is added to your available numbers.

7. You win when you have exactly one number left that matches the target.

For example, with target 50 and numbers [39, 66, 33, 13]:
**State 0** Target: 50
Operations: []
Available Numbers: [39, 66, 33, 13]
**Action 0** Operation: '39 + 13 = 52'
**State 1** (After performing 39 + 13 = 52)
Target: 50
Operations: ['39 + 13 = 52']
Available Numbers: [66, 33, 52]
**Action 1** Operation: '66 / 33 = 2'
**State 2** (After performing 66 / 33 = 2)
Target: 50
Operations: ['39 + 13 = 52', '66 / 33 = 2']
Available Numbers: [52, 2]
**Action 2** Operation: '52 - 2 = 50'
**State 3** (After performing 52 - 2 = 50)
Target: 50
Operations: ['39 + 13 = 52', '66 / 33 = 2', '52 - 2 = 50']
Available Numbers: [50]
**Game won!**

***Action Prior*** **System Instruction and User Request**

**System Instruction**

— — — — — — — — — — — **Insert game rules here** — — — — — — — — — — — — —

**Important considerations when assigning probabilities to operations:**

1. **Target Progress:** How much closer the operation gets to the target
   - Operations resulting in numbers exactly at or very close to target should receive higher scores
   - Operations creating useful intermediate numbers should be favored
2. **Number Creation:** The utility of the resulting number
   - Creating small, flexible numbers (1-10) can be valuable
   - Creating numbers that are factors of the target
   - Creating numbers that offer efficient pathways to the target
3. **Available Number Management:** How the operation affects the number pool
   - Operations that use less useful numbers while preserving useful ones
   - Operations that create a more workable set of available numbers
   - Avoiding operations that result in unusable large numbers
4. **Mathematical Strategy:** Using operations optimally
   - Using division to create useful small numbers
   - Using multiplication for larger adjustments toward the target
   - Using addition/subtraction for precise movements toward the target

Your task is to evaluate the possible actions in the current state, scoring them based on how likely they are to help you achieve the target value. The scores should form a probability distribution over the actions.

**Example State Sequence State 0** Target: 50
Operations: []
Available Numbers: [39, 66, 33, 13]
**Action 0** Operation: '39 + 13 = 52'
**State 1** (After performing 39 + 13 = 52)
Target: 50
Operations: ['39 + 13 = 52']
Available Numbers: [66, 33, 52]
**Action 1** Operation: '66 / 33 = 2'
**State 2** (After performing 66 / 33 = 2)
Target: 50
Operations: ['39 + 13 = 52', '66 / 33 = 2']
Available Numbers: [52, 2]
Example Possible Operations: {0: '52 + 2 = 54', 1: '52 - 2 = 50', 2: '52 * 2 = 104', 3: '52 / 2 = 26'}
**Example Final Answer**

{"operation_scores" : {"0" : 0.15, "1" : 0.35, "2" : 0.35, "3" : 0.15}}

**User Request**

**Current State and Action sequence** $\{current\_sequence\}$
Possible Operations: $\{action\_list\}$
What are the scores for each action/operation? Assign a probability to each possible operation based on how likely it is to lead to the target number.
Your response must include a valid JSON object, enclosed in a `boxed`, with an `operation_scores` field containing a dictionary mapping operation keys to scores, formatted as follows:

$$\{"operation\_scores" :< dictionary\_of\_scores >\}$$

Replace `<dictionary_of_scores>` with a dictionary mapping operation keys to scores that must sum to 1.0.

## *Estimate Node Value* System Instruction and User Request

### System Instruction

− − − − − − − − − − − − **Insert game rules here** − − − − − − − − − − − − − −

Important factors to consider when estimating state value:

1. **Proximity to Target:** How close the current numbers are to the target
   - States with numbers exactly equal to or close to the target are more valuable
   - States with numbers that can be easily combined to reach the target have higher value
2. **Available Number Quality:** How useful the remaining numbers are
   - Having small numbers (1-10) increases flexibility
   - Having numbers that are factors or multiples of target numbers is valuable
   - Having complementary numbers that work well together
3. **State Progress:** How much progress has been made
   - Number of operations performed so far
   - Reduction in the total number of available numbers
   - Quality of the operations performed so far
4. **Potential for Success:** Overall likelihood of reaching the target
   - Presence of clear pathways to the target
   - Absence of unusable or problematic numbers
   - Balance between large and small numbers

Your task is to estimate the value of the current state and possible operations by determining the likelihood of reaching the target number from it. The score should range from 0 to 1.

For example:

**Example State Sequence**

**State 0** Target: 50

Operations: []

Available Numbers: [39, 66, 33, 13]

**Action 0** Operation: '39 + 13 = 52'

**State 1** (After performing 39 + 13 = 52)

Target: 50

Operations: ['39 + 13 = 52']

Available Numbers: [66, 33, 52]

**Action 1** Operation: '66 / 33 = 2'

**State 2** (After performing 66 / 33 = 2)

Target: 50

Operations: ['39 + 13 = 52', '66 / 33 = 2']

Available Numbers: [52, 2]

Example Possible Operations: ['52 + 2 = 54', '52 - 2 = 50', '52 * 2 = 104', '52 / 2 = 26']

**Example Final Answer**

{"state_value_estimation" : 1.0}

**User Request**

**Current State and Action sequence** $\{current\_sequence\}$
Possible Operations: $\{action\_list\}$
Given the current state and the possible operations, estimate the value of the current state, ranging from 0-1, where 1 means it's certain to reach the target number and 0 means it's impossible.
Your response must include a valid JSON object, enclosed in a `boxed`, with a `state_value_estimation` field, formatted as follows:

$$\{\text{"state\_value\_estimation"} :< value >\}$$

Replace `<value>` with your estimated probability (between 0 and 1) of reaching the target from this state.

**_Move Values Estimation_ System Instruction and User Request**

**System Instruction**

— — — — — — — — — — — — **Insert game rules here** — — — — — — — — — — — — —

Important considerations when evaluating possible operations:

1. **Target Progress:** How much each operation moves toward the target
   - Operations that result in numbers close to the target
   - Operations that create useful intermediate numbers for future steps
2. **Number Creation:** The strategic value of the resulting number
   - Creating small, useful numbers (1-10) for fine adjustments
   - Creating numbers that are easily combinable with others
   - Creating numbers that are factors or related to the target
3. **Operation Strategy:** How the operation affects solution paths
   - Using division to create useful small numbers
   - Using multiplication to make larger jumps toward the target
   - Using addition/subtraction for precise adjustments
4. **Future Potential:** How an operation affects future possibilities
   - Operations that open up multiple future paths
   - Operations that eliminate problematic numbers
   - Operations that maintain flexibility in the number set

Your task is to evaluate each possible operation and assign a value between 0 and 1 to each, where 1 means the operation is extremely likely to lead to solving the puzzle and 0 means it's very unlikely to be helpful.
For example:
**Example State Sequence**
**State 0** Target: 50
Operations: []
Available Numbers: [39, 66, 33, 13]
**Action 0** Operation: '39 + 13 = 52'
**State 1** (After performing 39 + 13 = 52)
Target: 50
Operations: ['39 + 13 = 52']
Available Numbers: [66, 33, 52]
Example Possible Operations: {0: '52 + 66 = 118', 1: '52 - 33 = 19', 2: '66 - 33 = 33', 3: '66 / 33 = 2'}
**Example Final Answer**

{"operation_values" : {"0" : 0.3, "1" : 0.6, "2" : 0.5, "3" : 0.9}}

**User Request**

**Current State and Action sequence** $\{current\_sequence\}$
Possible Operations: $\{action\_list\}$
Evaluate each possible operation and assign a value between 0 and 1 to each, where 1 means the operation is extremely likely to lead to solving the puzzle and 0 means it's very unlikely to be helpful. Your response must include a valid JSON object, enclosed in a `boxed`, with an `operation_values` field containing a dictionary mapping operation keys to values between 0 and 1, formatted as follows:

$$\{\text{"operation\_values"} :< dictionary\_of\_values >\}$$

Replace `<dictionary_of_values>` with a dictionary mapping operation keys to values between 0 and 1.

*Exploration Decision* System Instruction and User Request

**System Instruction**

− − − − − − − − − − − − **Insert game rules here** − − − − − − − − − − − − −−

Important considerations when deciding whether to explore or continue:

1. **Current Path Quality:** How promising the current path appears
   - Presence of numbers close to the target
   - Quality and usefulness of available numbers
   - Clear pathways to reach the target from current numbers

2. **Current Path Issues:** Signs the current path may be problematic
   - Numbers far from the target with no clear way to combine them
   - Repeated patterns or circular operations
   - No beneficial operations remaining

3. **Exploration Value:** Potential benefit of trying other paths
   - Number of operations already performed on current path
   - Quality of alternative unexplored paths
   - Diminishing returns on current path

4. **Decision Confidence:** Certainty about current path viability
   - Clear evidence current path cannot reach target
   - Presence of obviously better unexplored paths
   - Risk assessment of continuing vs exploring

Your task is to decide whether to continue with the current state or to visit an unexplored state. Before deciding, carefully consider the current sequence of states and actions, as well as the available operations. Only choose to explore if you are certain that the current path cannot reach the target number and that switching to a new path is the best use of time.

For example:

**Example State and Action sequence**

**State 0** Target: 50
Operations: []
Available Numbers: [39, 66, 33, 13]
**Action 0** Operation: '39 + 13 = 52'
**State 1** (After performing 39 + 13 = 52)
Target: 50
Operations: ['39 + 13 = 52']
Available Numbers: [66, 33, 52]
**Action 1** Operation: '66 / 33 = 2'
**State 2** (After performing 66 / 33 = 2)
Target: 50
Operations: ['39 + 13 = 52', '66 / 33 = 2']
Available Numbers: [52, 2]
Example Possible Operations: {0: '52 + 2 = 54', 1: '52 - 2 = 50', 2: '52 * 2 = 104', 3: '52 / 2 = 26'}
**Example Final Answer**

{"explore" : false}

**User Request**

**Current State and Action sequence** $\{current\_sequence\}$

Possible Operations: $\{action\_list\}$

Consider the current sequence of states and actions and the available operations. Reason through your options step by step and determine whether continuing with the current state or exploring a new state is the most optimal decision.

Your response must include a valid JSON object, enclosed in a `boxed`, with an `explore` field, where the value must be either true (to explore a new state) or false (to continue with the current state), formatted as follows:

$$\boxed{\{\text{"explore"} :< boolean >\}}$$

Replace `<boolean>` with either true or false.

## E.2  SUDOKU

**Sudoku Game Rules**

You are helping solve Sudoku puzzles using a tree-based search approach. Sudoku is a puzzle where you fill a grid with numbers 1 through $\{grid\_size\}$ so that each row, column, and box has no repeated numbers.

For this $\{grid\_size\} \times \{grid\_size\}$ Sudoku grid, the boxes are $\{box\_width\} \times \{box\_height\}$ in size. Each row, column, and box must contain all numbers from 1 to $\{grid\_size\}$ without repetition. This means:

1. Each row must contain each number from 1 to $\{grid\_size\}$ exactly once

2. Each column must contain each number from 1 to $\{grid\_size\}$ exactly once

3. Each $\{box\_width\} \times \{box\_height\}$ box must contain each number from 1 to $\{grid\_size\}$ exactly once

These constraints create a logical puzzle where placing a number in a cell immediately restricts what numbers can be placed in other cells in the same row, column, and box.

**Board Structure:**

- The Sudoku board is a $\{grid\_size\} \times \{grid\_size\}$ grid divided into $\{box\_width\} \times \{box\_height\}$ boxes

- Rows are numbered 0 to $\{grid\_size\_minus\_one\}$ from top to bottom

- Columns are numbered 0 to $\{grid\_size\_minus\_one\}$ from left to right

- Each cell is identified by its (row, column) coordinates

- Empty cells appear as periods (.) in the board representation

- Board state is represented as a nested list where `board[row][column]` gives the value at that position

When solving a Sudoku puzzle, we explore different possible number placements. Each step involves selecting an empty cell and placing a valid number in it. As we make selections, the set of valid moves for remaining cells may change.

### *Action Prior* System Instruction and User Request

**System Instruction**

─────────────────────────────────────

− − − − − − − − − − − − **Insert game rules here** − − − − − − − − − − − − − −

─────────────────────────────────────

**Important considerations when evaluating possible actions:**

1. How actions might create naked singles or hidden singles in other cells
2. Actions targeting cells with few remaining alternatives
3. How actions may constrain multiple other cells simultaneously
4. How actions contribute to a balanced distribution of numbers across the board
5. Whether actions might lead to contradictions or cells with no legal moves

Your task is to evaluate the possible actions in the current state, scoring them based on how likely they are to help solve the Sudoku puzzle. The scores should form a probability distribution over the actions (sum to 1.0) and be returned as a dictionary mapping action indices to scores.
**Example** $\{grid\_size\} \times \{grid\_size\}$ **Sudoku Board**
$\{example\_board\}$
**Example Possible Actions**
$\{example\_prior\_actions\}$
**Example Final Answer**

$\{\text{"operation\_scores"} : \{example\_operation\_scores\}\}$

**User Request**

**Current** $\{grid\_size\} \times \{grid\_size\}$ **Sudoku Board**
$\{current\_board\}$
**Possible Actions**
$\{action\_list\}$
Evaluate each action based on how it creates constraints, identifies singles, minimizes branching, and maintains a balanced distribution of numbers as described in your instructions.
Assign a probability to each possible action based on how likely it is to lead to a solution of the Sudoku puzzle. The scores should sum to 1.0, representing a probability distribution over the actions.
Your response must include a valid JSON object, enclosed in a boxed, with an `operation_scores` field containing a dictionary mapping action indices to scores, formatted as follows:

$\{\text{"operation\_scores"} :< \text{dictionary\_of\_scores} >\}$

Replace `<dictionary_of_scores>` with a dictionary mapping action indices to scores that **MUST** sum to 1.0.

## *Node Value* System Instruction and User Request

### System Instruction

—————————— **Insert game rules here** ——————————

**Important considerations when estimating the value of a board state:**

*1. Factors that may indicate higher likelihood of success:*

- The number of cells with few possible remaining values
- Whether all cells have at least one possible legal value
- How close rows, columns, and boxes are to completion
- The presence of obvious next moves such as naked or hidden singles

*2. Factors that may indicate lower likelihood of success:*

- The presence of cells with zero possible legal values (contradictions)
- Many cells having numerous possible values (high uncertainty)
- Limited constraints between remaining empty cells
- Patterns that typically lead to unsolvable states

Your task is to estimate the value of the current board state by determining the likelihood of solving the puzzle from this position. The score should range from 0 to 1.

**Example** $\{grid\_size\} \times \{grid\_size\}$ **Sudoku Board**

$\{example\_board\}$

**Example Possible Actions**

$\{example\_value\_actions\}$

**Example Final Answer**

$$\{\text{"state\_value\_estimation"} : 0.75\}$$

### User Request

**Current** $\{grid\_size\} \times \{grid\_size\}$ **Sudoku Board**

$\{current\_board\}$

**Possible Actions**

$\{action\_list\}$

Given the current board state and the possible actions, estimate the value of the current state. Consider factors like the number of cells with few possible values, whether there are contradictions, and whether there are obvious next moves as described in your instructions.

Provide a value ranging from 0–1, where 1 means it's certain to reach a solution and 0 means it's impossible.

Your response must include a valid JSON object, enclosed in a `boxed`, with a `state_value_estimation` field, formatted as follows:

$$\{\text{"state\_value\_estimation"} :< \text{value} >\}$$

Replace `<value>` with your estimated probability (between 0 and 1) of solving the puzzle from this state.

## *Explore Decision* System Instruction and User Request

### System Instruction

―――――――――――― **Insert game rules here** ―――――――――――――

**Important considerations when determining whether to continue with the current board state or explore a new state:**

1. The presence of naked singles or hidden singles in the current board state

2. Whether the current board state contains contradictions or cells with no valid moves

3. The level of certainty in the remaining cells (many vs. few possible values)

4. Whether the board shows signs of making progress or appears to be in a deadlock

Your task is to decide whether to continue with the current board state or to visit an unexplored board state. Before deciding, carefully consider the current board and the available actions. Only choose to explore if you are certain that the current board state cannot lead to a solution and that switching to a new board state is the best use of time.
**Example** $\{grid\_size\} \times \{grid\_size\}$ **Sudoku Board** $\{example\_board\}$
**Example Possible Moves** $\{example\_explore\_actions\}$
**Example Final Answer**

$$\{\text{"explore"} : false\}$$

### User Request

**Current** $\{grid\_size\} \times \{grid\_size\}$ **Sudoku Board** $\{current\_board\}$
**Possible Moves** $\{empty\_cells\}$
Consider the current board state and the available actions. Evaluate whether the current state has promising moves like naked singles or hidden singles, or if it shows signs of contradictions or deadlocks as described in your instructions.
Reason through your options step by step and determine whether continuing with the current state or exploring a new state is the most optimal decision.
Respond with true if you should explore a new board state, or false if you should continue with the current one.
Your response must include a valid JSON object, enclosed in a `boxed`, with an `explore` field, where the value must be either true (to explore a new board state) or false (to continue with the current board state), formatted as follows:

$$\{\text{"explore"} :< \text{boolean} >\}$$

Replace `<boolean>` with either true or false.

*Move Value Estimation* System Instruction and User Request

**System Instruction**

− − − − − − − − − − − − − **Insert game rules here** − − − − − − − − − − − − − −

**Important considerations when evaluating possible moves:**

1. **Constraint Propagation:** How each move affects future possibilities
   - Whether the move creates naked singles or hidden singles
   - How the move constrains other cells in the same row, column, and box
2. **Strategic Value:** The quality of the move in solving the puzzle
   - Whether the move targets cells with few remaining possibilities
   - Whether the move maintains flexibility in other cells
   - Whether the move creates a balanced distribution of numbers
3. **Future Impact:** How the move affects future solving paths
   - Whether the move opens up multiple solving techniques
   - Whether the move might lead to contradictions
   - Whether the move maintains good solving options

Your task is to evaluate each possible move and assign a value between 0 and 1 to each, where 1 means the move is extremely likely to lead to solving the puzzle and 0 means it's very unlikely to be helpful.
**Example** $\{grid\_size\} \times \{grid\_size\}$ **Sudoku Board** $\{example\_board\}$
**Example Possible Moves** $\{example\_moves\}$
**Example Final Answer**

$$\{\text{"move\_values"} : \{\text{"0"} : 0.8, \text{"1"} : 0.5, \text{"2"} : 0.3, \dots \}\}$$

**User Request**

**Current** $\{grid\_size\} \times \{grid\_size\}$ **Sudoku Board** $\{current\_board\}$
**Possible Moves** $\{moves\_list\}$
Evaluate each possible move and assign a value between 0 and 1 to each, where 1 means the move is extremely likely to lead to solving the puzzle and 0 means it's very unlikely to be helpful.
Your response must include a valid JSON object, enclosed in a `boxed`, with a `move_values` field containing a dictionary mapping move indices to values between 0 and 1, formatted as follows:

$$\{\text{"move\_values"} :< dictionary\_of\_values >\}$$

Replace `<dictionary_of_values>` with a dictionary mapping move indices to values between 0 and 1.

# F  PRELIMINARY INVESTIGATION: MCTS EXPLORATION CONSTANT

We performed a hyperparameter sweep over different exploration constants $C \in \{0.5, 1.0, 2.5\}$. Due to computational constraints, we limited this sweep to the three Countdown variants and the simpler Sudoku

variant, using GPT-4o as the underlying model. As shown in Figures 4, 5, and 6, the setting $c = 2.5$ consistently underperforms, while $c = 0.5$ and $c = 1.0$ perform similarly, with $c = 0.5$ slightly outperforming $c = 1.0$ in Countdown (difficulty 5). The largest performance gap appears in the Sudoku (4x4) task (Figure 7), where $c = 0.5$ significantly outperforms higher values. This is likely due to Sudoku's deeper solution space, where higher $c$-values lead to over-exploration. The overall trend is further confirmed by the performance profiles in Figures 2 and 3, which show $c = 0.5$ achieving the best trade-off between performance and efficiency. Based on these results, we adopt $c = 0.5$ as the default value in subsequent experiments.

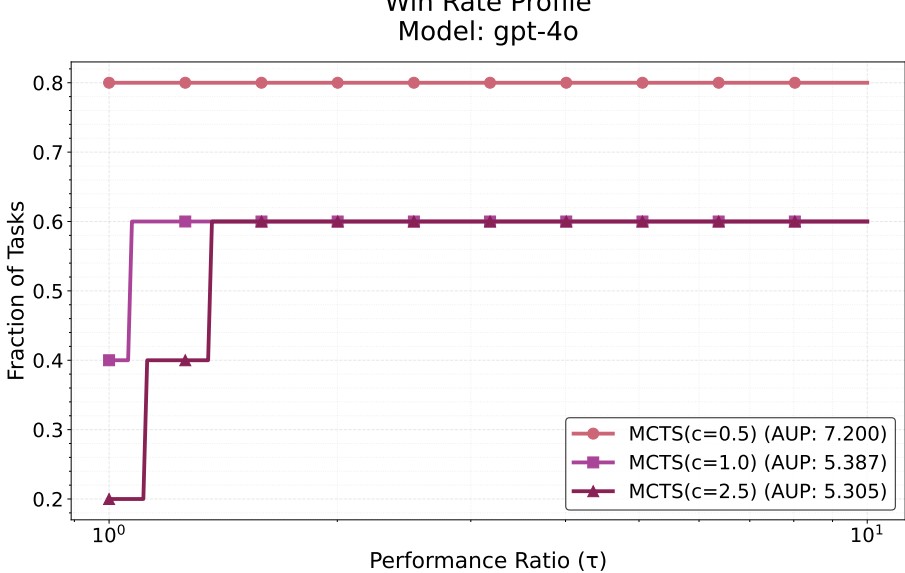

Figure 2: Performance profiles of MCTS across different exploration constants ($c \in \{0.5, 1.0, 2.5\}$), evaluated using WinRate across all tasks with GPT-4o. The profiles illustrate the proportion of tasks where each $c$ value is within a given performance ratio of the best. Area Under the Profile (AUP) is displayed for each curve. Notably, $c = 0.5$ achieves the highest AUP, indicating superior overall performance.

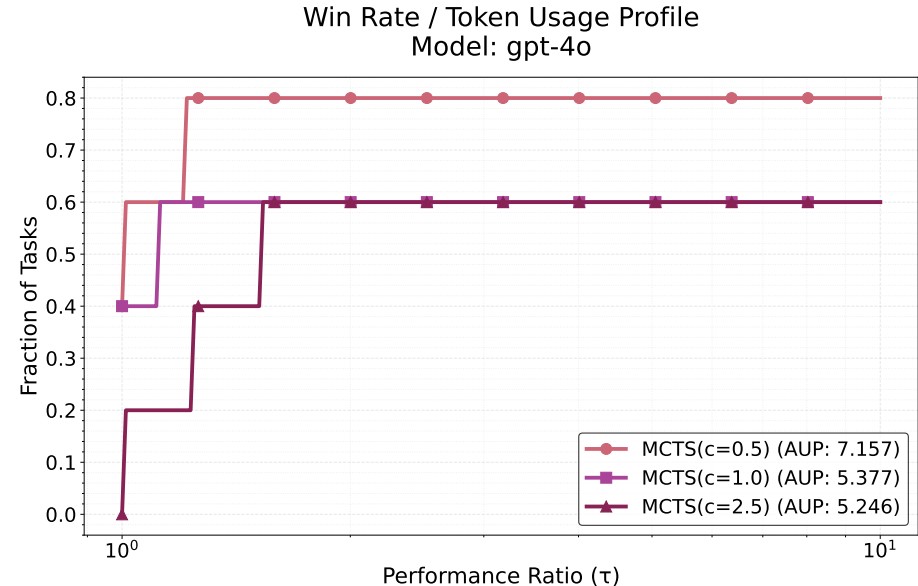

Figure 3: Performance profiles of MCTS across different exploration constants ($c \in \{0.5, 1.0, 2.5\}$), evaluated using WinRate per token ratio (efficiency) across all tasks with GPT-4o. The profiles indicate the proportion of tasks where each $c$ value achieves a given efficiency ratio relative to the best. Area Under the Profile (AUP) is shown for each curve. As with overall WinRate, $c = 0.5$ yields the highest AUP, demonstrating superior efficiency.

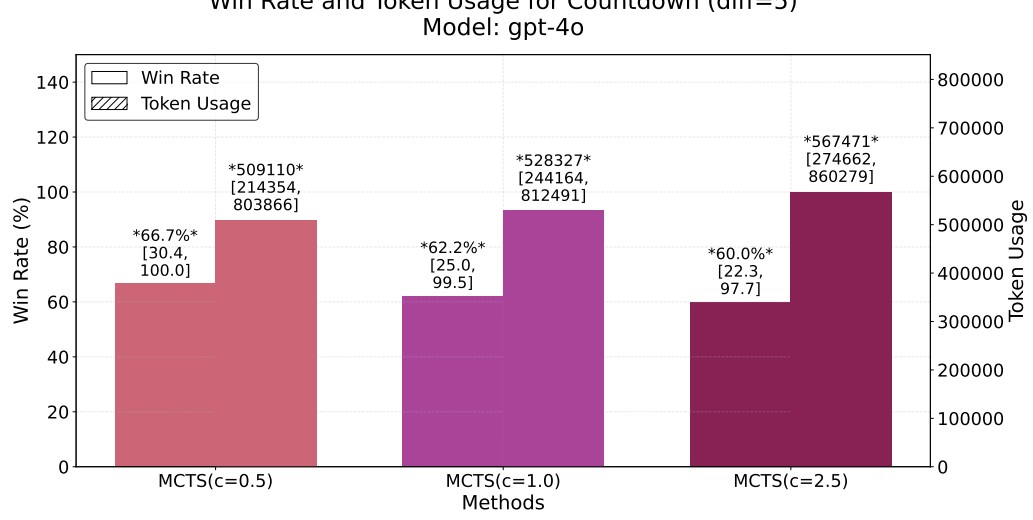

Figure 4: WinRate and token usage for MCTS on the Countdown task (difficulty 3) using GPT-4o. Both metrics are reported across different exploration constants ($c = 0.5, 1.0, 2.5$), with all configurations successfully solving all instances. Notably, $c = 0.5$ uses the most tokens. Values in "*" denote the mean, and square brackets "[ ]" represent the 95% confidence interval.

Figure 5: WinRate and token usage for MCTS on the Countdown task (difficulty 5) using GPT-4o. Results are shown for exploration constants $c = 0.5, 1.0,$ and $2.5$. See that $c = 0.5$ achieves the best WinRate while also using the fewest tokens on average. Values in "*" denote the mean, and square brackets "[ ]" represent the 95% confidence interval.

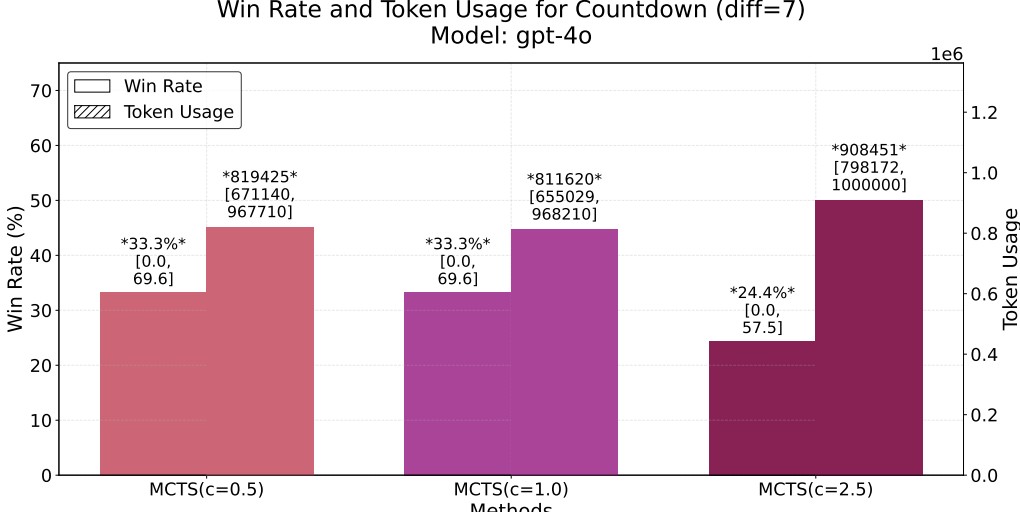

Figure 6: WinRate and token usage for MCTS on the Countdown task (difficulty 7) using GPT-4o. Results are shown for exploration constants $c = 0.5$, $1.0$, and $2.5$. Both $c = 0.5$ and $c = 1.0$ achieve equal win rates, with $c = 1.0$ using marginally fewer tokens on average. Values in "*" denote the mean, and square brackets "[ ]" represent the 95% confidence interval.

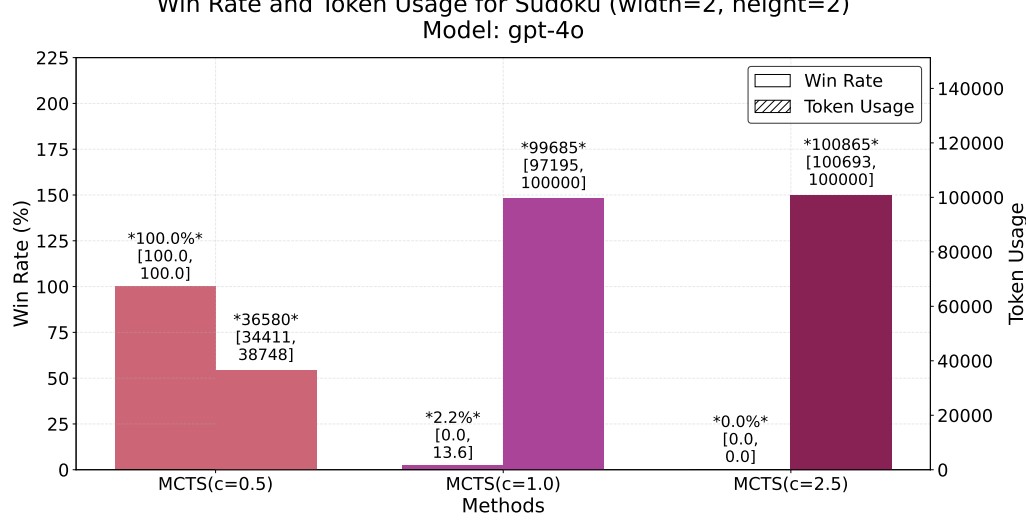

Figure 7: WinRate and token usage for MCTS on the Sudoku (4×4) task using GPT-4o. Results are shown for exploration constants $c = 0.5$, $1.0$, and $2.5$. Only $c = 0.5$ successfully solves all games, and it does so with significantly lower token usage compared to the other $c$ values, which struggle to solve any. Values in "*" denote the mean, and square brackets "[ ]" represent the 95% confidence interval.

# G ADDITIONAL EXPERIMENT RESULTS

Below, we present detailed experimental results across all Countdown and Sudoku variants. The subsections are organized as follows: performance profiles G.1, Countdown results G.2, Sudoku results G.3, cumulative wins G.4, and tree size analyses G.5.

## G.1 PERFORMANCE PROFILES

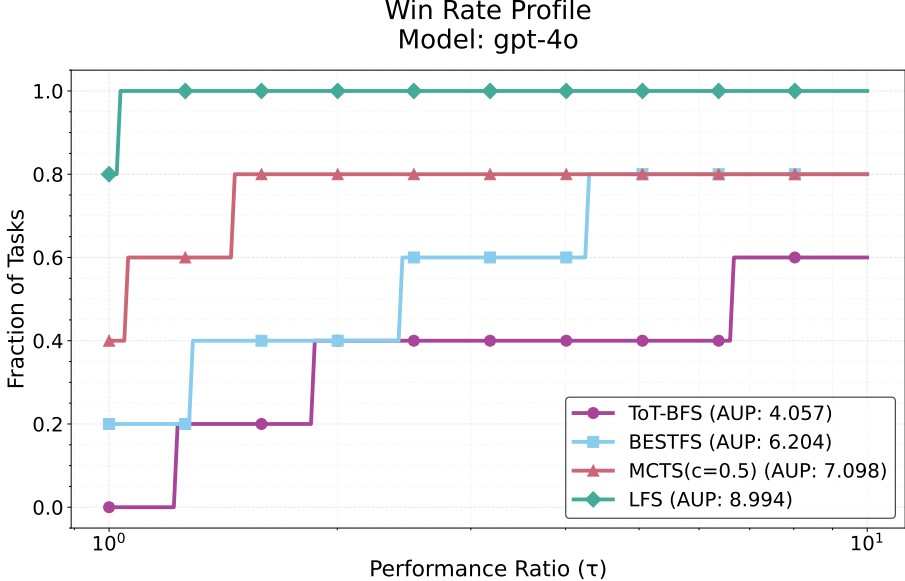

Figure 8: Performance profiles (WinRate) across all variants of Countdown and Sudoku tasks for methods ToT-BFS, BestFS, MCTS, and LFS, evaluated with GPT-4o. **LFS achieves the highest Area Under Profile (AUP) value**, indicating superior overall WinRate.

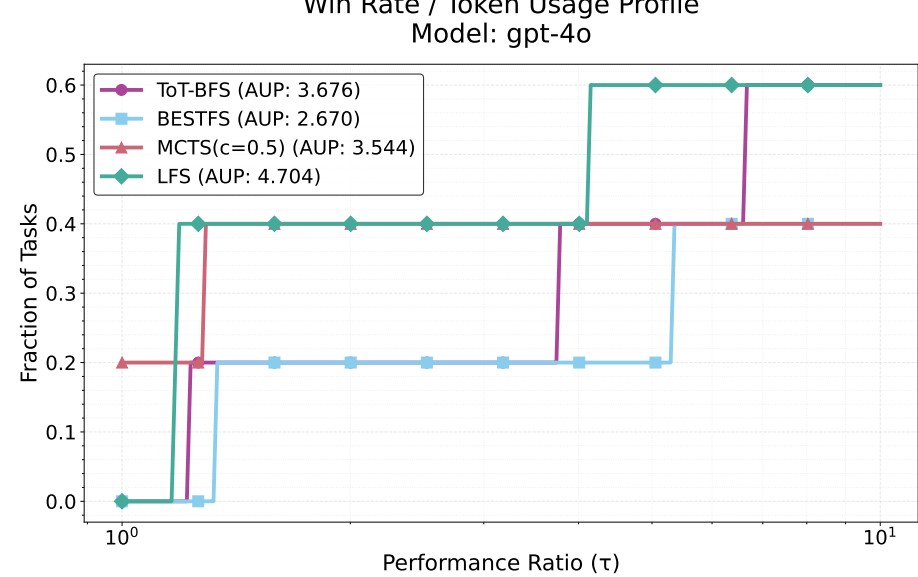

Figure 9: Performance profiles (WinRate per Token Ratio) across all variants of Countdown and Sudoku tasks for methods ToT-BFS, BestFS, MCTS, and LFS, evaluated with GPT-4o. Among these, **LFS achieves the highest Area Under Profile (AUP) value**, indicating it provides the best balance between WinRate and token efficiency.

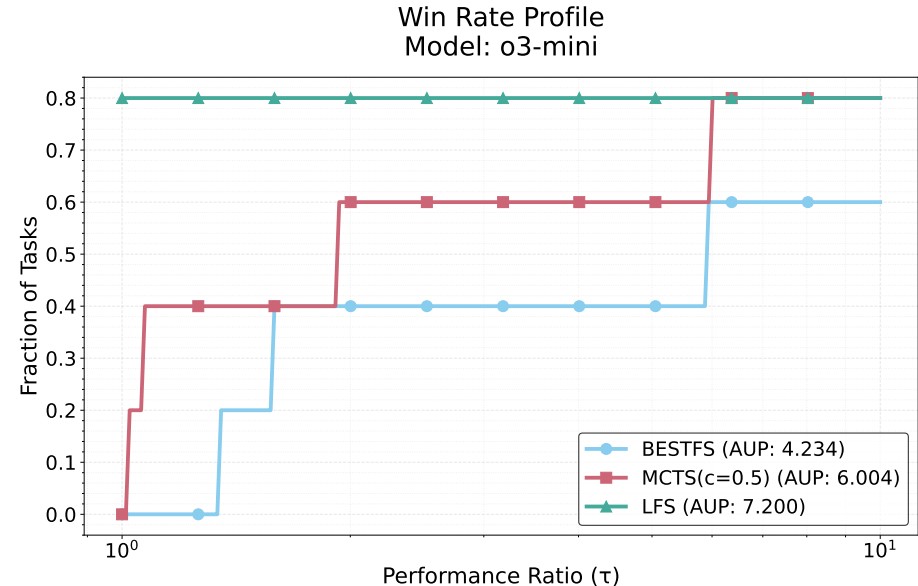

Figure 10: Performance profiles (WinRate) across all variants of Countdown and Sudoku tasks for methods BestFS, MCTS, and LFS, evaluated with o3-mini. Among these, **LFS achieves the highest Area Under Profile (AUP) value**, demonstrating superior overall performance.

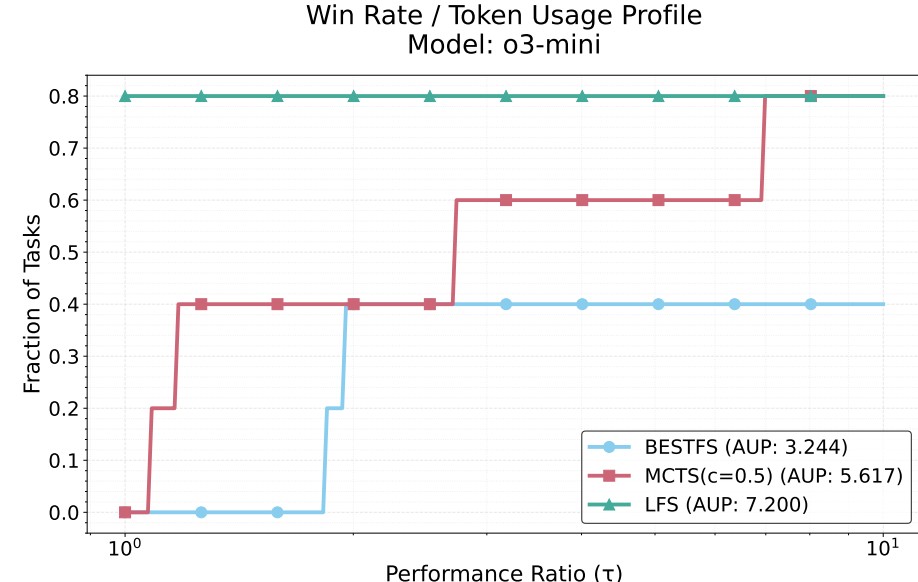

Figure 11: Performance profiles (WinRate per Token Ratio) across all variants of Countdown and Sudoku tasks for methods BestFS, MCTS, and LFS, evaluated with o3-mini. **LFS achieves the highest Area Under Profile (AUP) value**, indicating the best efficiency-performance trade-off among the methods.

fewe

## G.2 COUNTDOWN RESULTS

### GPT-4O RESULTS

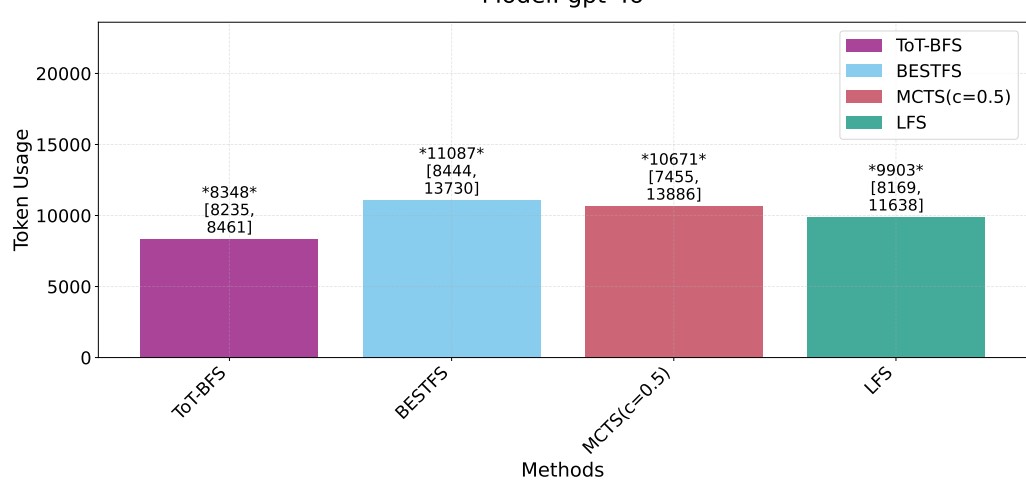

(a) Win rates for difficulty 3.

(b) Token usage for difficulty 3.

Figure 12: WinRate and token usage for different methods (ToT-BFS, BestFS, MCTS, and LFS) on the Countdown task (difficulty 3) using GPT-4o. **(a)** WinRate; **(b)** Token Usage. ToT-BFS was the only method that did not solve all instances, while the other three methods successfully solved all tasks. Among these three, LFS used the fewest tokens, indicating the best efficiency. Values in "*" denote the mean, and square brackets "[ ]" represent the 95% confidence interval.

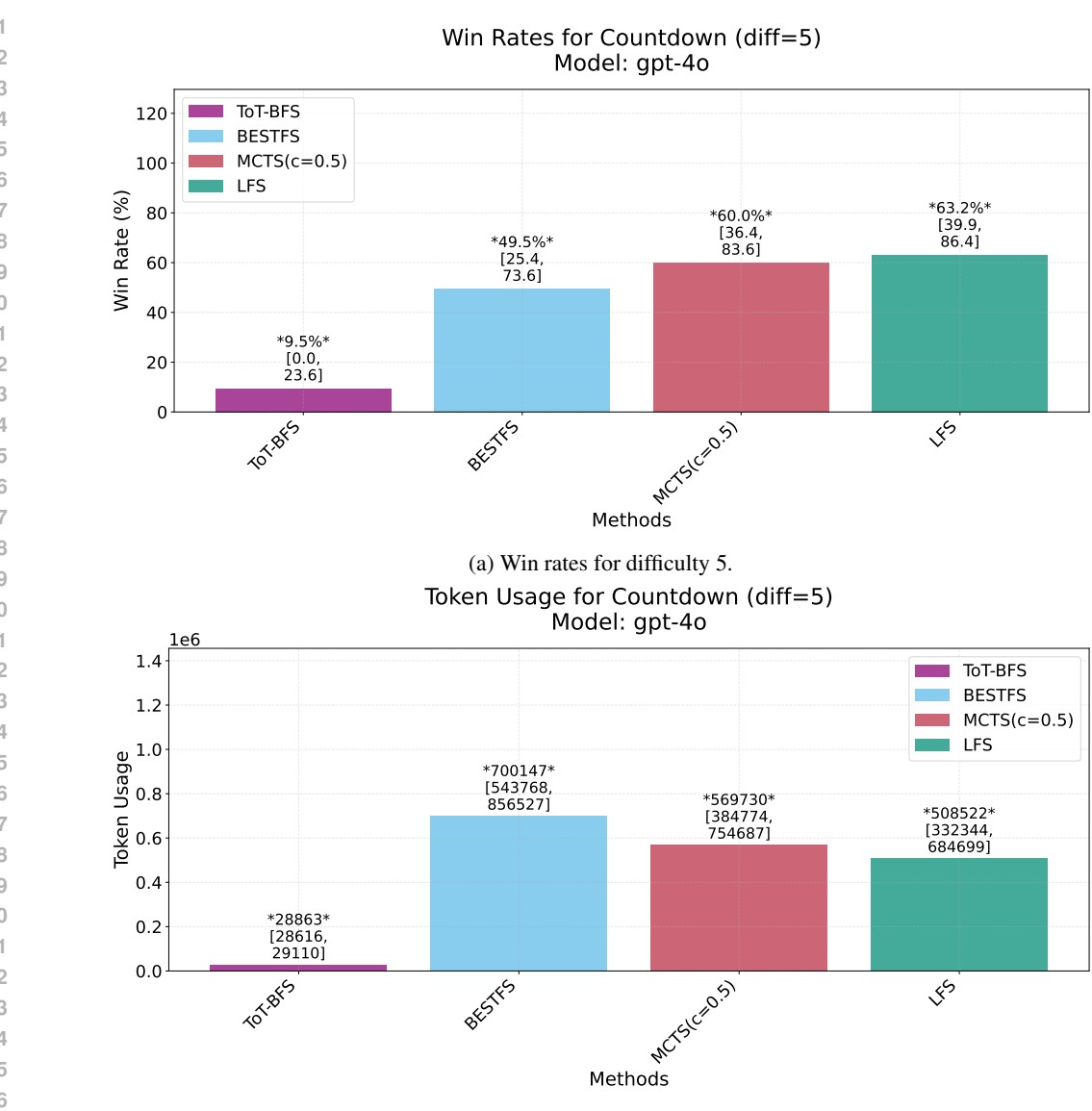

(a) Win rates for difficulty 5.

(b) Token usage for difficulty 5.

Figure 13: WinRate and token usage for different methods (ToT-BFS, BestFS, MCTS, and LFS) on the Countdown task (difficulty 5) using GPT-4o. **(a)** WinRate; **(b)** Token Usage. LFS marginally outperforms the next best method, MCTS, while also using fewer tokens, indicating both higher effectiveness and efficiency. Values in "∗" denote the mean, and square brackets "[ ]" represent the 95% confidence interval.

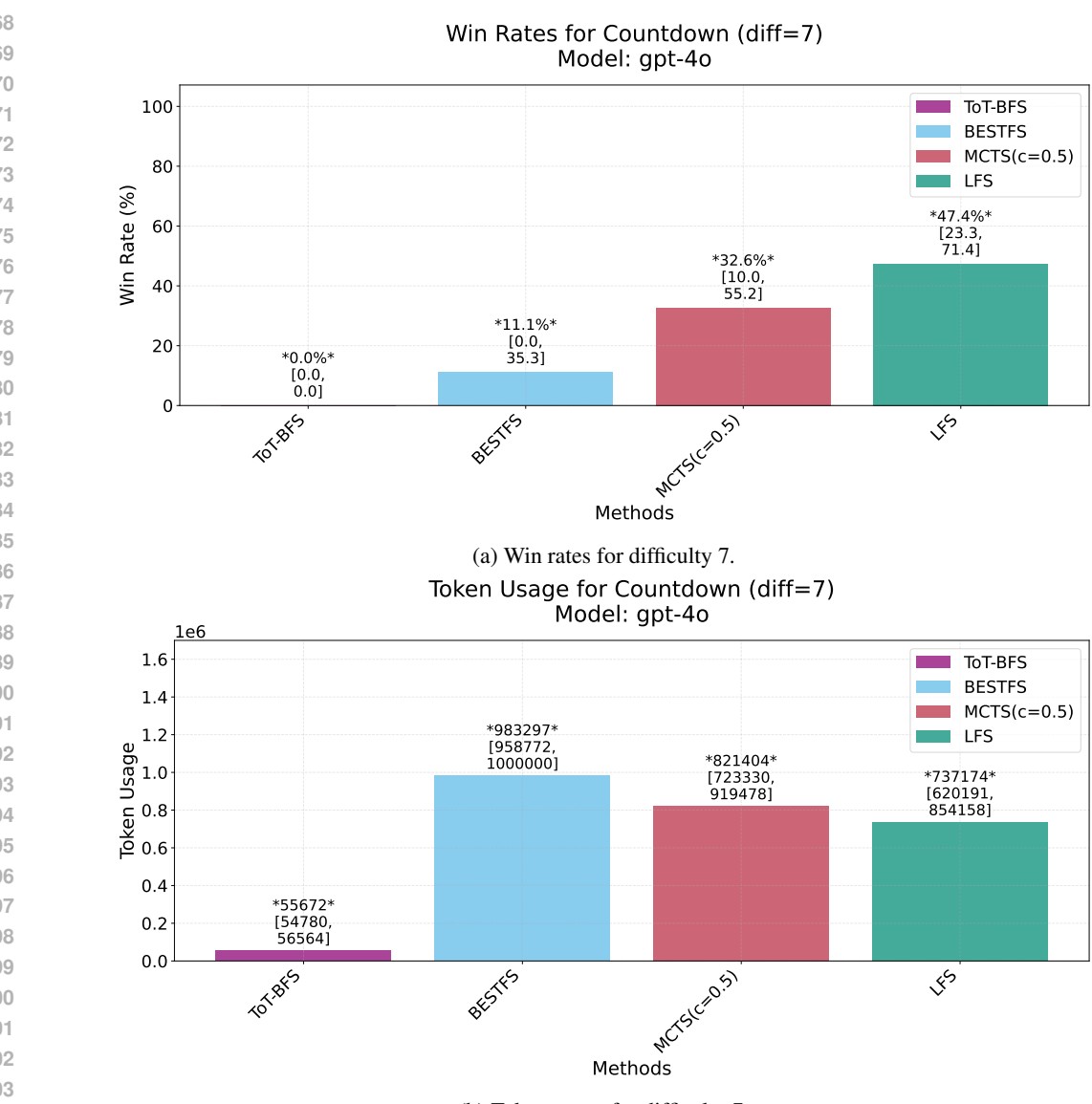

(a) Win rates for difficulty 7.

(b) Token usage for difficulty 7.

Figure 14: WinRate and Token Usage for different methods (ToT-BFS, BestFS, MCTS, and LFS) on the Countdown task (difficulty 7) using GPT-4o. **(a)** WinRate; **(b)** Token Usage. The performance gap between MCTS and LFS widens as difficulty increases, with LFS maintaining higher efficiency by using fewer tokens. Values in "*" denote the mean, and square brackets "[ ]" represent the 95% confidence interval.

O3-MINI RESULTS

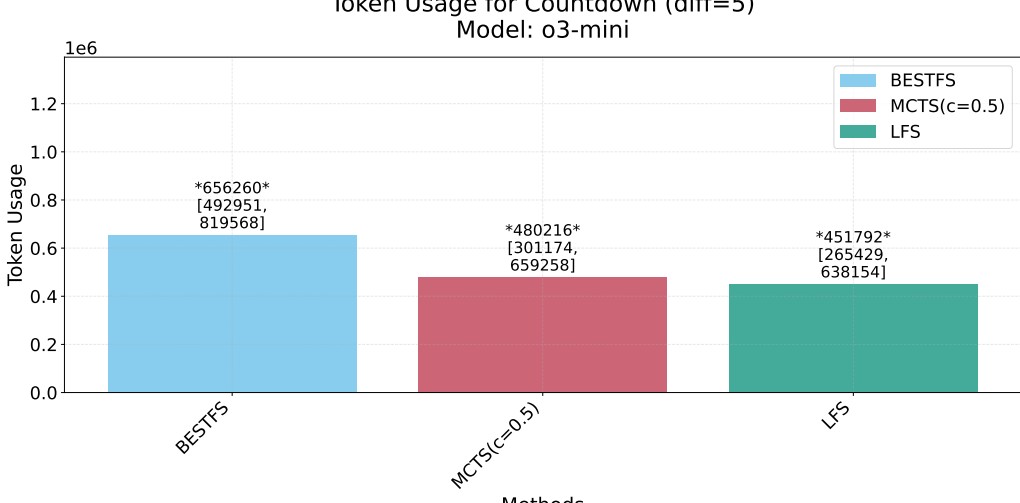

(a) Win rates for Countdown (difficulty 5).

(b) Token usage for Countdown (difficulty 5).

Figure 15: WinRate and Token Usage for different methods (BestFS, MCTS, and LFS) on the Countdown task (difficulty 5) using o3-mini. (a) WinRate; (b) Token Usage. The performance trends closely mirror those observed with GPT-4o: LFS marginally outperforms MCTS while also using fewer tokens, indicating stronger efficiency. Values in "*" denote the mean, and square brackets "[ ]" represent the 95% confidence interval.

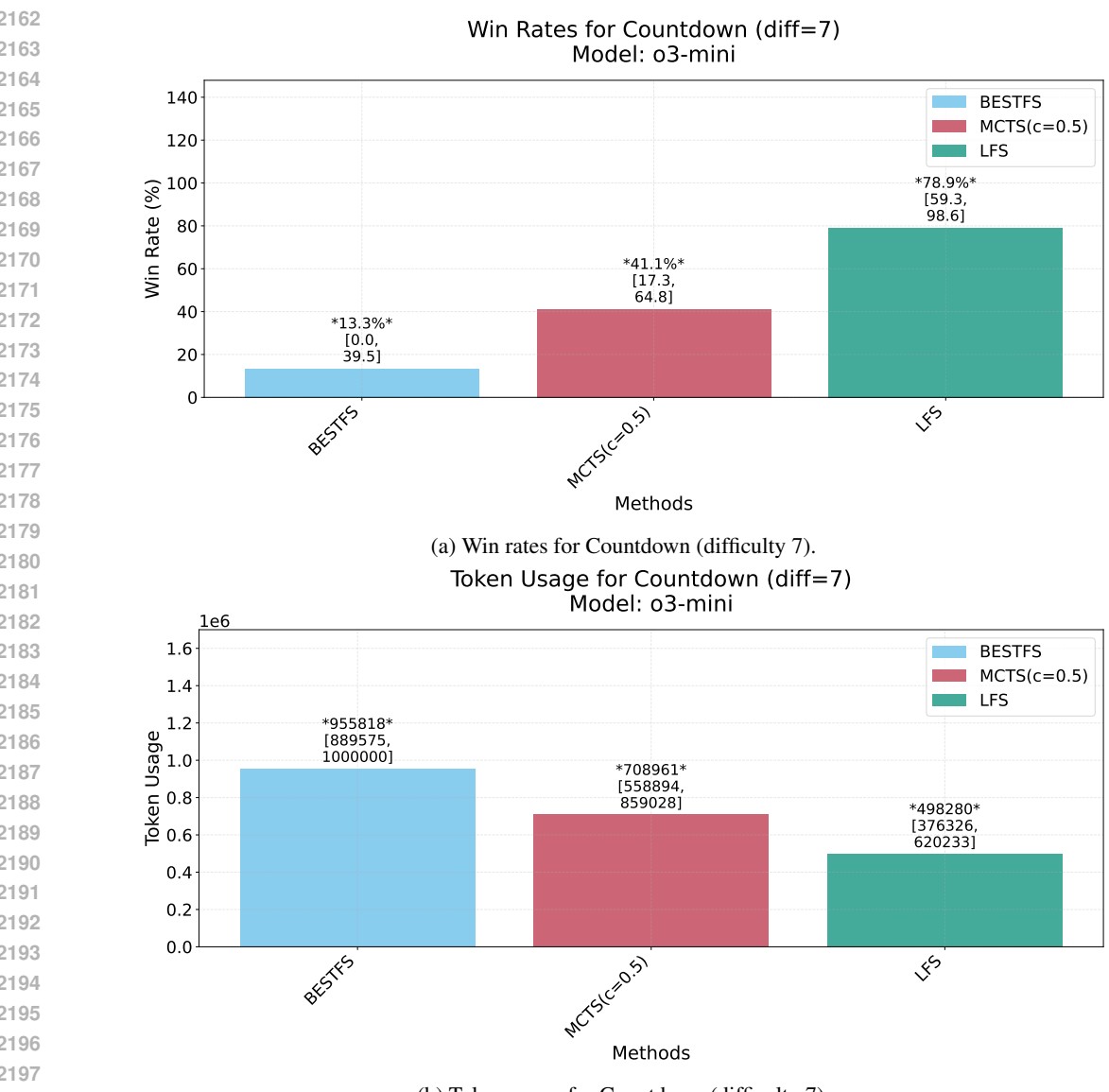

(a) Win rates for Countdown (difficulty 7).

(b) Token usage for Countdown (difficulty 7).

Figure 16: WinRate and Token Usage for different methods (BestFS, MCTS, and LFS) on the Countdown task (difficulty 7) using `o3-mini`. **(a)** WinRate; **(b)** Token Usage. The performance gap between MCTS and LFS widens as task difficulty increases, mirroring results with GPT-4o, with LFS maintaining higher efficiency through lower token usage. Values in "*" denote the mean, and square brackets "[ ]" represent the 95% confidence interval.

### G.3 Sudoku Results

**GPT-4o Results**

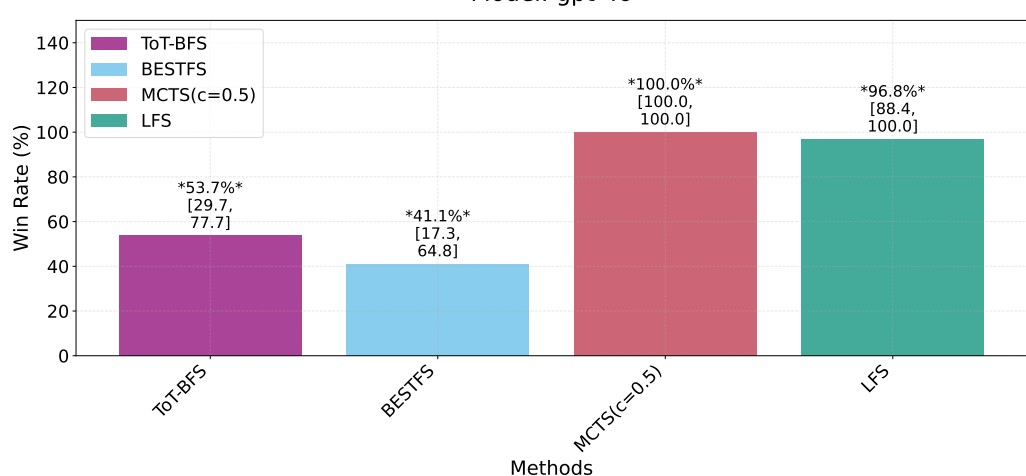

(a) Win rates for Sudoku $4 \times 4$.

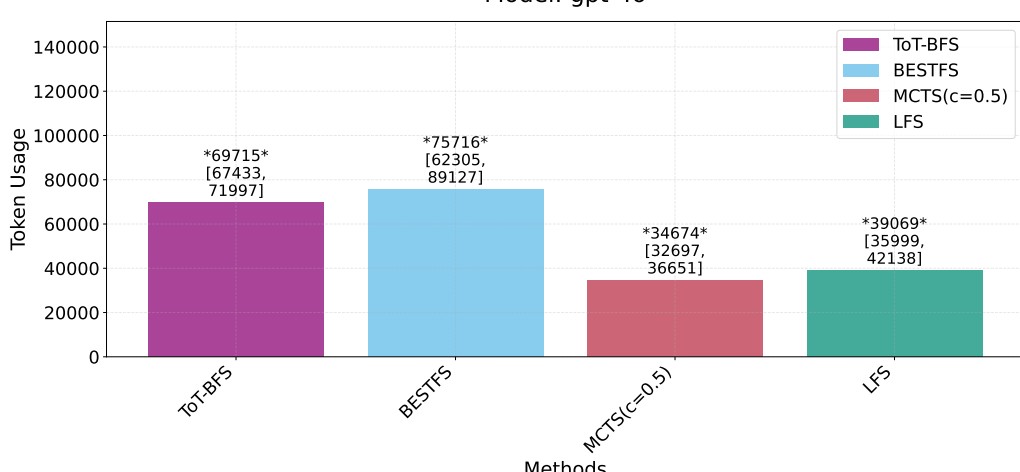

(b) Token usage for Sudoku $4 \times 4$.

Figure 17: WinRate and Token Usage on the Sudoku $4 \times 4$ task using `GPT-4o`. **(a)** WinRate; **(b)** Token Usage. Results are shown for ToT-BFS, BestFS, MCTS, and LFS. MCTS marginally outperforms LFS in both WinRate and token efficiency, while ToT-BFS and BestFS lag significantly behind. Values in "*" denote the mean, and square brackets "[ ]" represent the 95% confidence interval.

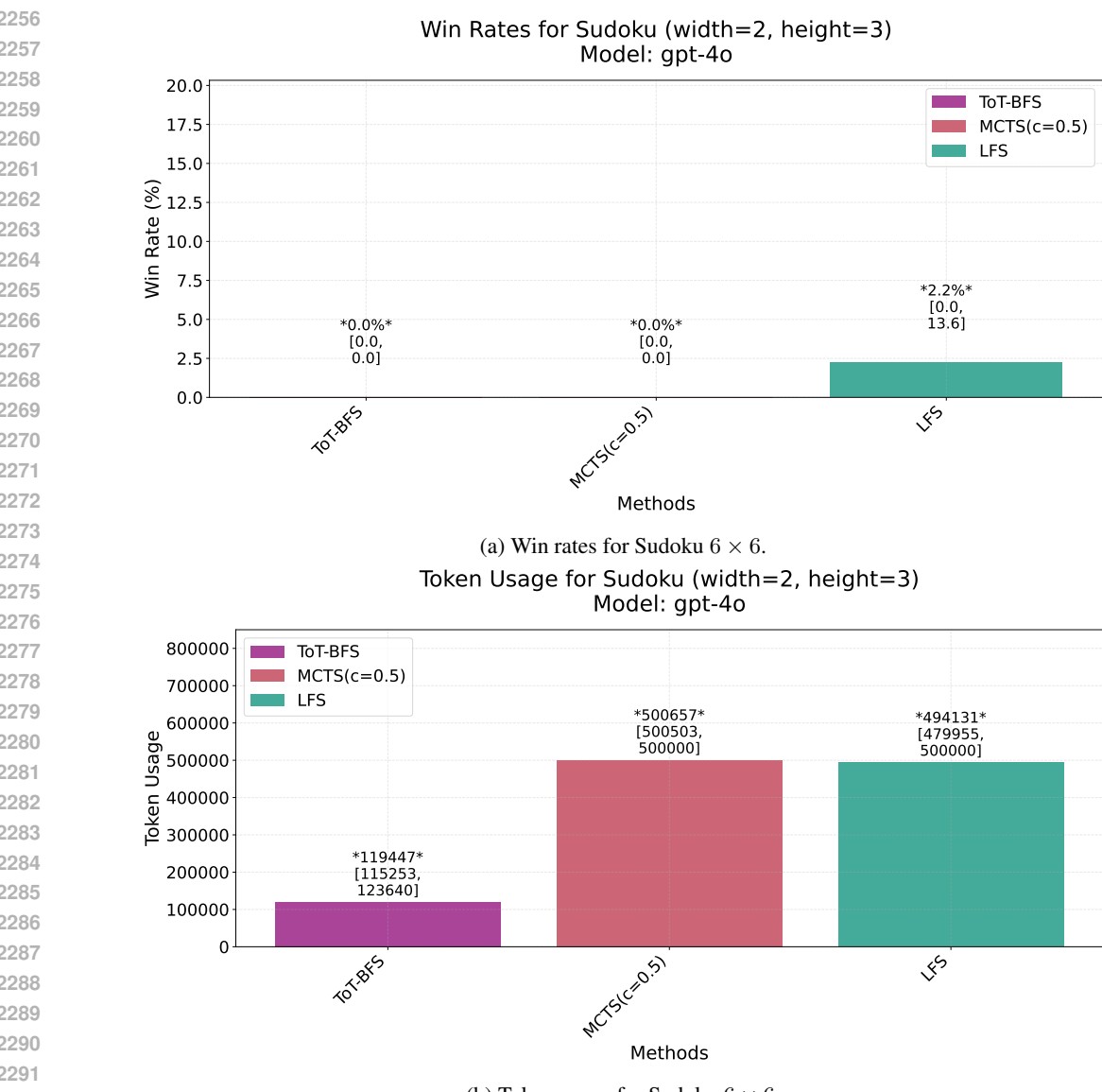

(a) Win rates for Sudoku $6 \times 6$.

(b) Token usage for Sudoku $6 \times 6$.

Figure 18: WinRate and Token Usage on the Sudoku $6 \times 6$ task using `GPT-4o`. **(a)** WinRate; **(b)** Token Usage. Results are shown for ToT-BFS, MCTS, and LFS. All methods fail to solve any instances, except LFS, which successfully solves a single game. Despite the overall difficulty. Values in "*" denote the mean, and square brackets "[ ]" represent the 95% confidence interval.

O3-MINI RESULTS

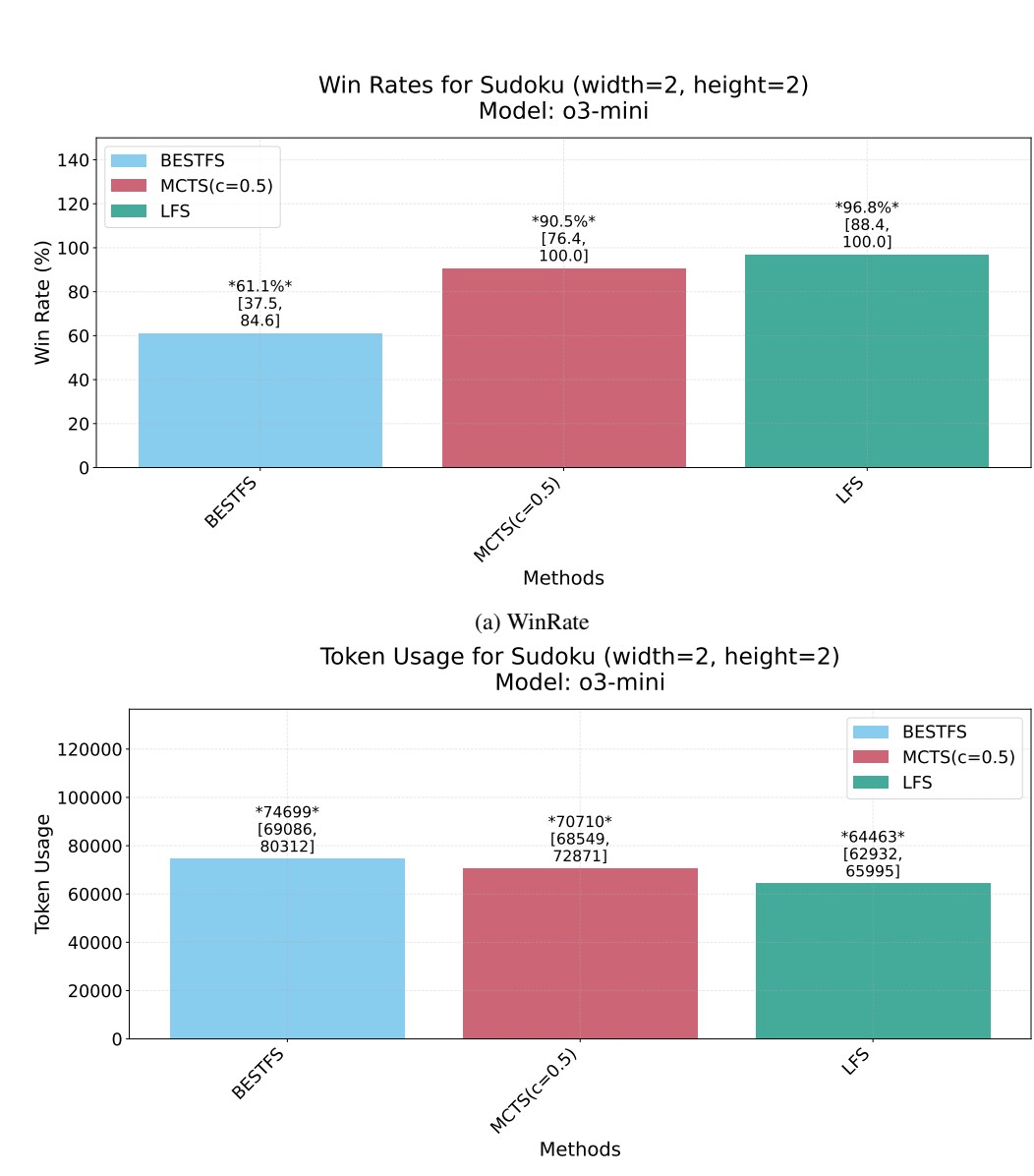

(a) WinRate

(b) Token Usage

Figure 19: WinRate and Token Usage on the Sudoku $4 \times 4$ task using o3-mini. **(a)** WinRate; **(b)** Token Usage. Results are shown for BestFS, MCTS, and LFS. Unlike the GPT-4o setting, LFS now outperforms MCTS in both WinRate and token efficiency, highlighting that our method scales more effectively with stronger models. Values in "*" denote the mean, and square brackets "[ ]" represent the 95% confidence interval.

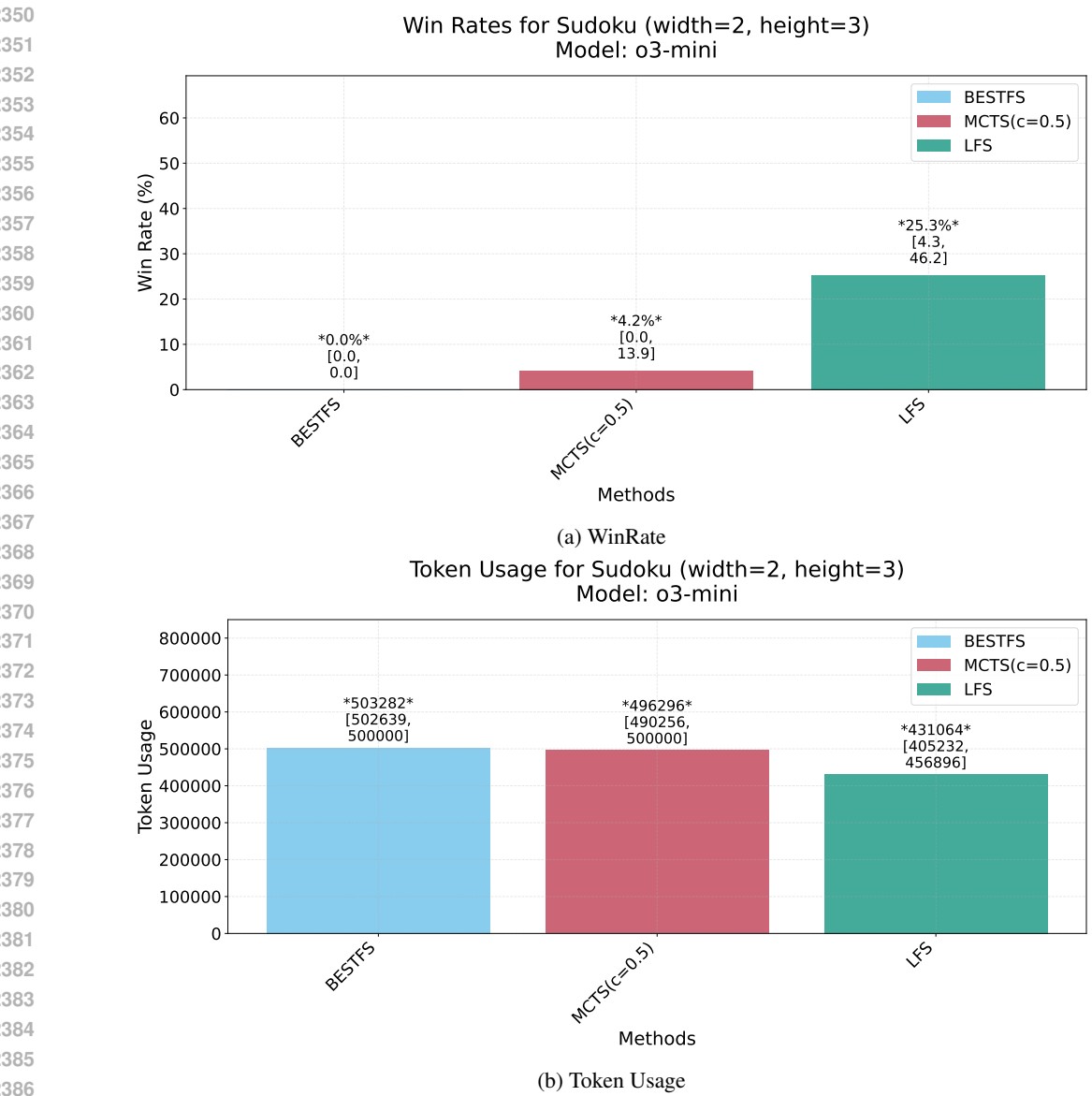

(a) WinRate

(b) Token Usage

Figure 20: WinRate and Token Usage on the Sudoku $6 \times 6$ task using `o3-mini`. **(a)** WinRate; **(b)** Token Usage. Results are shown for BestFS, MCTS, and LFS. The trend from the $4 \times 4$ variant continues, with LFS significantly outperforming MCTS in both accuracy and token efficiency. This indicates that LFS scales more effectively with stronger models and handles more difficult tasks more robustly. Values in "*" denote the mean, and square brackets "[ ]" represent the 95% confidence interval.

## G.4 CUMULATIVE WINS

We provide detailed results illustrating the cumulative wins achieved by different methods as the token budget increases for both Countdown and Sudoku games. As shown in Figures 21a and 21b, the total number of Countdown games won steadily rises with higher token usage, with LFS clearly outperforming the next best method, MCTS. This performance gap is especially pronounced for the stronger o3-mini model (Figure 21b), indicating that LFS scales more effectively with model strength. Although compute limitations prevented testing at larger token budgets, the current trend suggests this gap would continue to widen. A similar but less prominent pattern can be observed for Sudoku (Figures 21c and 21d), where WinRate saturation on simpler Sudoku variants and overall lower performance on harder variants temper the advantage.

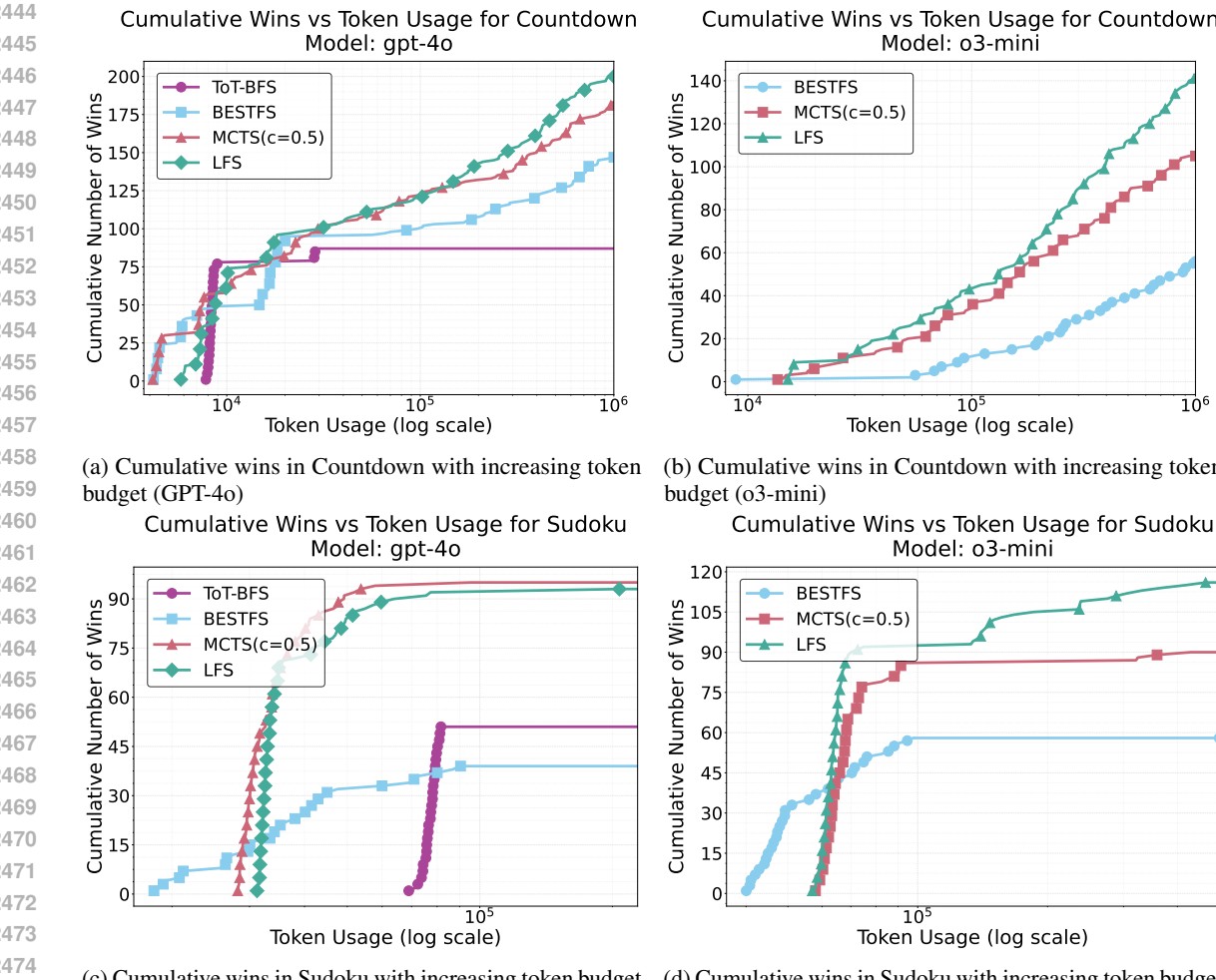

(a) Cumulative wins in Countdown with increasing token budget (GPT-4o)

(b) Cumulative wins in Countdown with increasing token budget (o3-mini)

(c) Cumulative wins in Sudoku with increasing token budget (GPT-4o).

(d) Cumulative wins in Sudoku with increasing token budget (o3-mini).

Figure 21: Cumulative wins across varying token budgets for Countdown and Sudoku games using different methods. Panels (a) and (b) show Countdown results for GPT-4o and o3-mini models respectively, highlighting the superior scalability of LFS over MCTS, particularly with the stronger model. Panels (c) and (d) display cumulative Sudoku wins, where the performance gap is less pronounced due to WinRate saturation and increased task difficulty.

## G.5 TREE SIZE

We report the average tree sizes generated by each method across different levels of difficulty for both the Countdown and Sudoku domains, using the GPT-4o and o3-mini models. In the Countdown setting, we observe that LFS consistently constructs smaller or equal-sized trees compared to MCTS. A similar pattern emerges in the Sudoku tasks, across both the $4 \times 4$ and $6 \times 6$ grid configurations. These results illustrate the efficiency of LFS's guided exploration strategy, which avoids the over-exploration characteristic of MCTS, and maintains performance even as problem complexity increases.

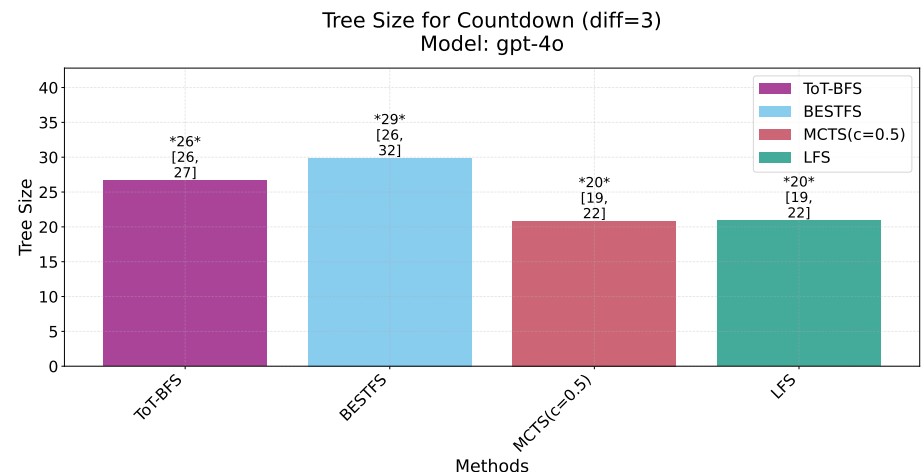

Figure 22: Average tree size for Countdown (difficulty 3) using GPT-4o.

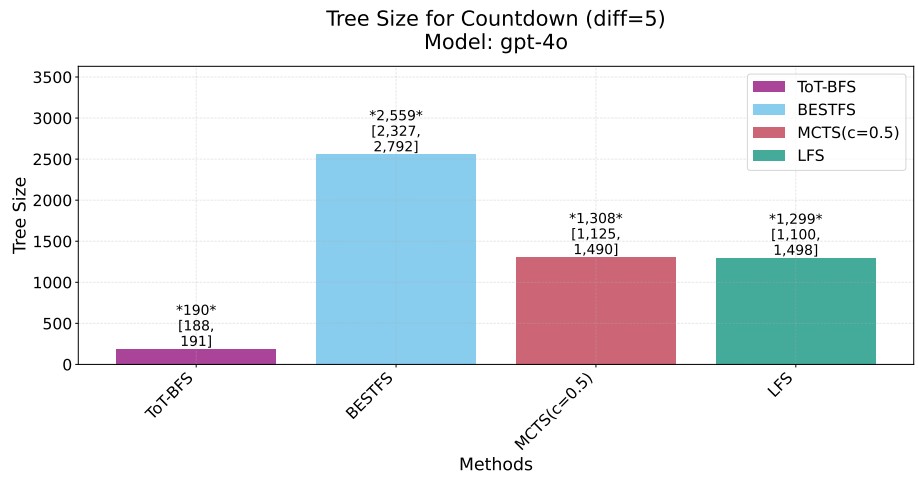

Figure 23: Average tree size for Countdown (difficulty 5) using GPT-4o.

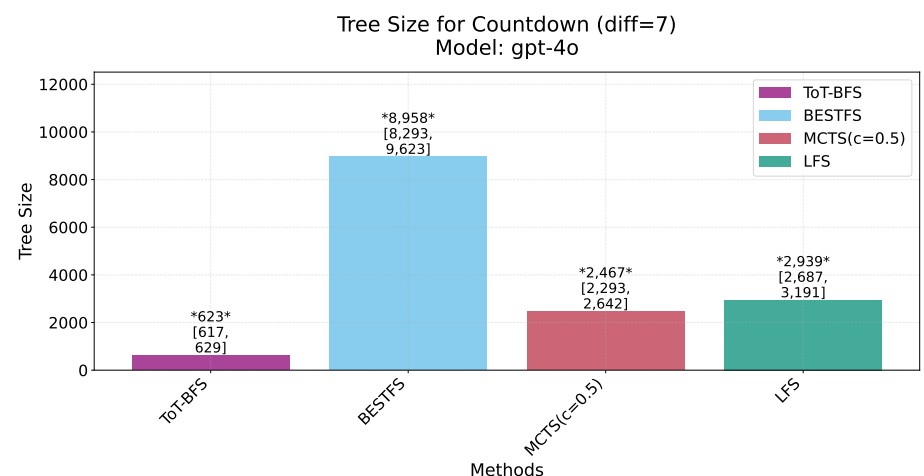

Figure 24: Average tree size for Countdown (difficulty 7) using GPT-4o.

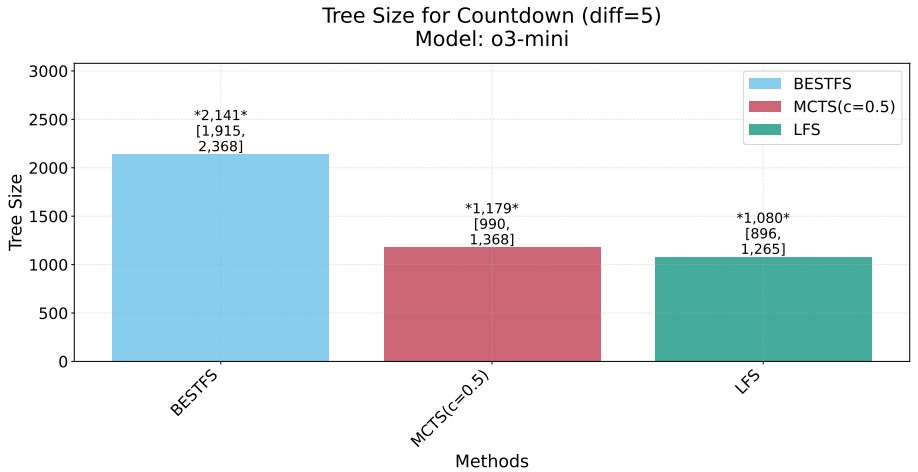

Figure 25: Average tree size for Countdown (difficulty 5) using o3-mini.

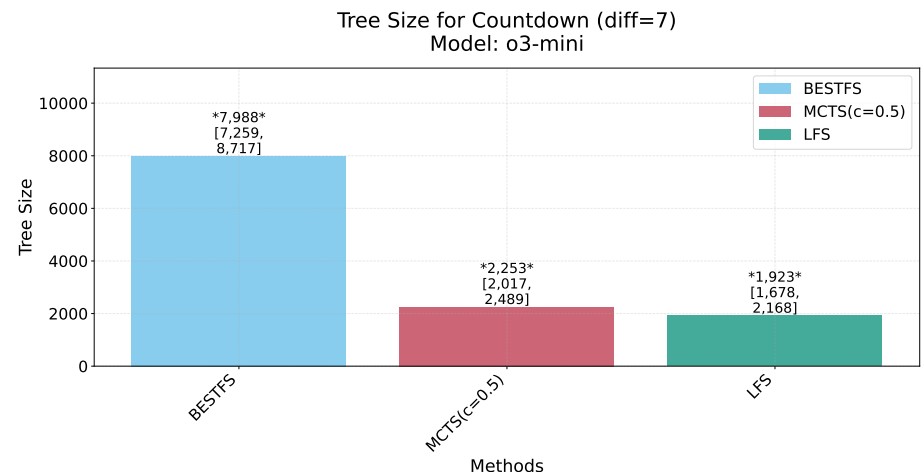

Figure 26: Average tree size for Countdown (difficulty 7) using o3-mini.

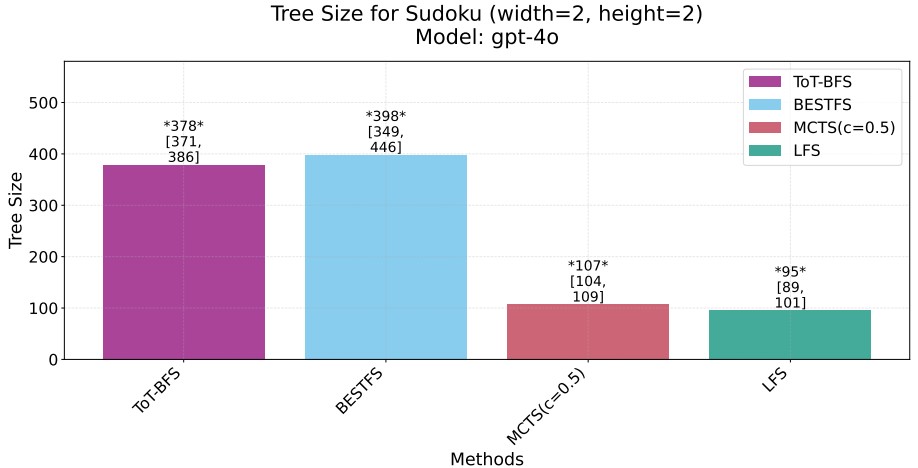

Figure 27: Average tree size for Sudoku ($2 \times 2$) using GPT-4o.

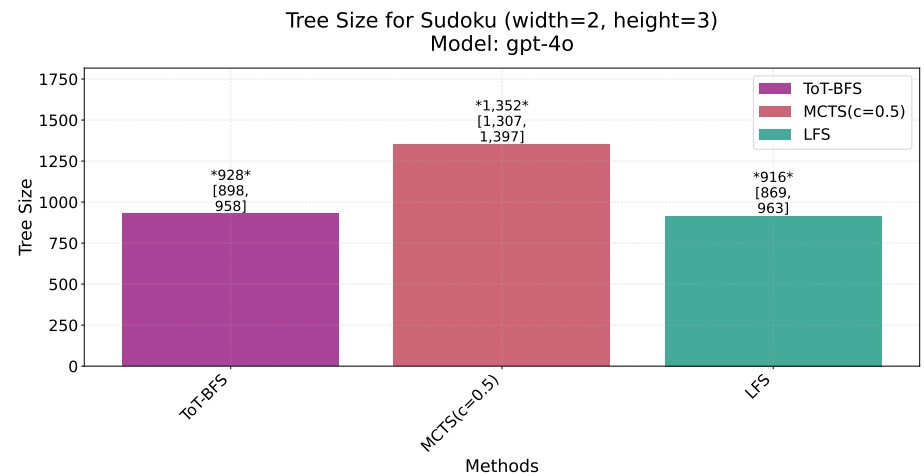

Figure 28: Average tree size for Sudoku ($2 \times 3$) using GPT-4o.

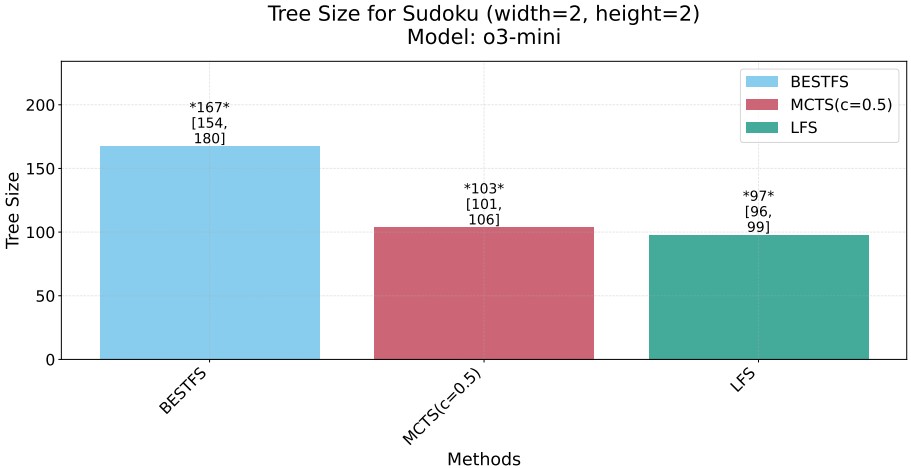

Figure 29: Average tree size for Sudoku ($2 \times 2$) using o3-mini.

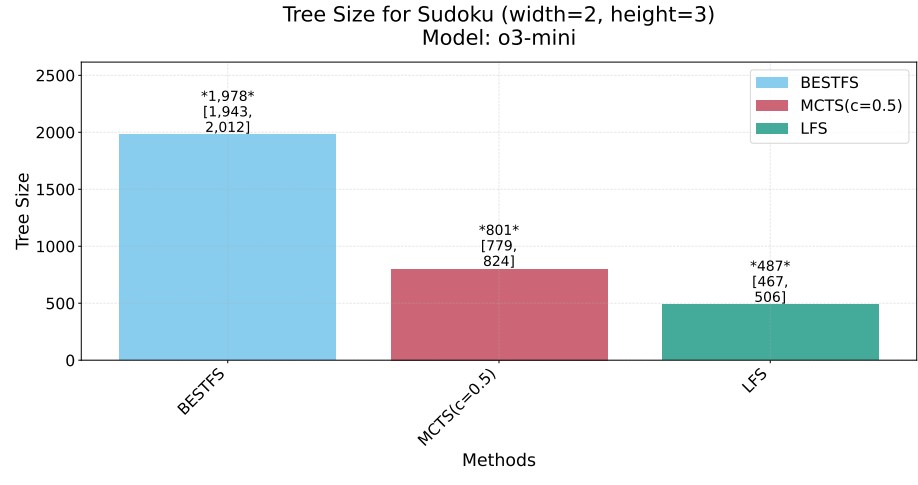

Figure 30: Average tree size for Sudoku ($2 \times 3$) using o3-mini.

## H  EXAMPLE TREES

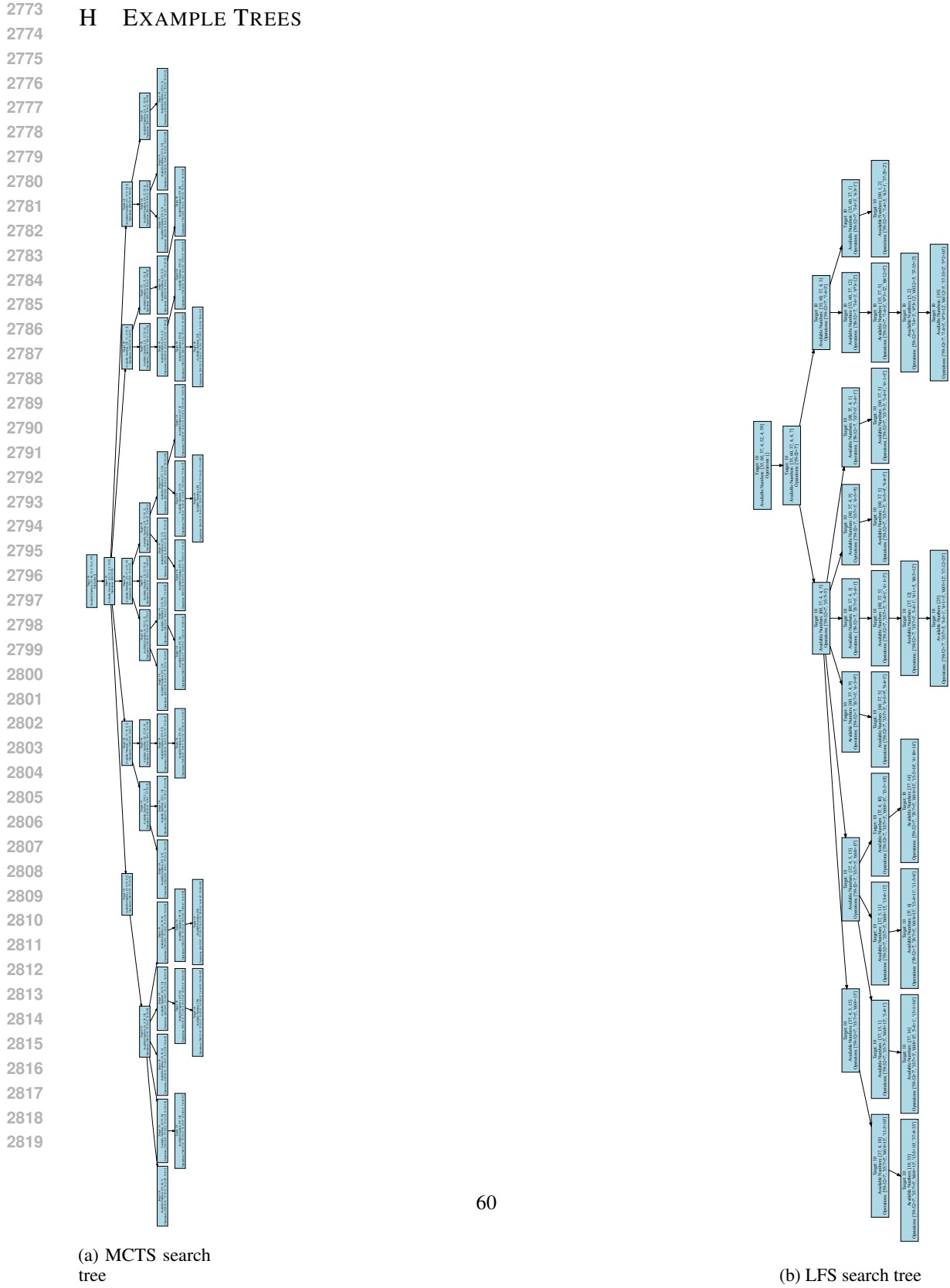

(a) MCTS search
tree

(b) LFS search tree

Figure 31: Example search trees generated for a Countdown game (difficulty = 7) using (a) Monte Carlo
Tree Search (MCTS) and (b) Limited-Depth Forward Search (LFS). The MCTS tree is noticeably wider,
illustrating its tendency for over-exploration compared to the more focused LFS tree.

Figure 31 shows example search trees generated for a Countdown game with difficulty level 7. Subfigure (a) depicts the tree produced by MCTS, while subfigure (b) shows the tree from LFS. Notice that the MCTS tree is considerably wider, reflecting its tendency to over-explore the search space. In contrast, the LFS tree is more focused and narrower, indicating a more targeted exploration strategy. This comparison highlights the differences in exploration behaviour between the two methods on the same problem instance.