# OpenReview forum: "LLM-First Search: Self-Guided Exploration of the Solution Space"
_ICLR.cc/2026/Conference — Submitted to ICLR 2026_

### Official Review · Reviewer_FYPb · 2025-10-24

**Soundness:** 2
**Presentation:** 3
**Contribution:** 3
**Rating:** 6
**Confidence:** 3

**Summary:**

The paper proposes LLM-First Search (LFS), a test-time LLM agentic framework where the LLM explore different decision nodes in a tree-based manner similar to prior works, but prompts LLM itself to give scores to every expandable node in the tree and decide whether to explore or exploit. If the LLM decides to explore and jump from the current reasoning trace to another trace, the LLM will switch to the most promising alternative node retrieved by a priority queue; otherwise, the LLM will continue diving into the current node and update the priority queue. The proposed method outperforms several baselines on two tasks, which are Sudoku and Countdown.

**Strengths:**

1. The idea is simple, intuitive and clearly conveyed: to remove handcrafted rule for exploration-exploitation and let LLM take charge of which node to expand during tree search.

2. The related work seems to be rather complete, which discusses different LLM test-time reasoning framework in details and clearly stated the difference between the proposed method and prior works.

3. The paper has a very detailed appendix, which greatly increases the reproducibility of the paper by providing all the prompts and implementation details. The appendix also provides detailed experiment results which clearly show the advantage of LFS.

**Weaknesses:**

1. As mentioned in the limitation section, the proposed method highly relies on the LLM's base ability as the exploration-exploitation tradeoff is made by the LLM itself. However, if future long-context, thinking state-of-the-art models are strong enough, they may inherently possess the ability to jump between different branches of thoughts without adopting LFS (e.g. many papers [1] mention crucial tokens such as "wait" or "however" in solving complicated math problems, which can be seen as a variant of LFS).

2. Following 1, the motivation of getting rid of "fixed handcrafted exploration" becomes less convincing as LFS also adopts somewhat handcrafted way of expanding nodes (priority queue and the explicit, explore-exploit judgment), and MCTS-based methods also involves LLM-guided evaluation of scores as mentioned in Alg. 4. The question is: since every method contains more or less degree of "fixed handcrafted exploration" compared to expecting the "wait" or "however" token from LLM itself, for state-of-the-art models, where should we stop and say "this level of exploration-exploitation strategy contains neither too much nor too little handcrafted prior?"

**Minor Weakness**

1. In line 652, "LLM-First- Search" -> "LLM-First-Search"

2. In line 222, "Three-of-Thoughts" -> "Tree-of-Thoughts"

3. The formatting in page 27 is strange; it only has one line saying "Exploration Decision System Instruction and User Request".


**References**

[1] S. Wang et al. Beyond the 80/20 Rule: High-Entropy Minority Tokens Drive Effective Reinforcement Learning for LLM Reasoning, 2025.

**Questions:**

I have several questions:

1. The authors mention that the MCTS tree is noticably wider and over-explores. This is somewhat counterintuitive, as MCTS's rollout always goes on until the leaf / terminal (see Alg. 4 line 6-9), while in LFS the LLM can jump to another branch in the middle of a rollout (see Alg. 1); i.e, MCTS's rollouts are on average deeper. Can the author explain this?

2. Following 1, the authors also mention that the result shows that MCTS with a value of $C=0.5$ (i.e. fewest exploration) works better than higher $C$. The question is: will MCTS's performance grow further if $C$ is further decreased to, e.g., 0.1 or 0.2? Is 0.5 still "too high an exploration constant" as suggested in line 85?

3. As there are many possible future states, the priority queue could save many unexplored nodes. However, the LLM ususally gives a discrete scale (e.g. 0.05, 0.1, 0.15, ..., 1) instead of a continuous value between 0 and 1. Does LFS often encounter nodes with the same weights in the priority queue? If so, how does LFS deal with this?

**Details Of Ethics Concerns:**

The paper violates the double-blind policy. In page 10 of the paper, the authors include acknowledgement, which says: "This work was supported by the UK Engineering and Physical Sciences Research Council (EPSRC) under grant number EP/S021566/1", which gives away the identity of the authors.

---

> ### Author Response · Authors · 2025-11-15
> **Response to Reviewer FYPb**
>
> ### 1. Reliance on the LLM’s Base Ability
>
> We thank the reviewer for raising this important point. It is true that LFS leverages the LLM’s underlying reasoning ability to make exploration–exploitation decisions. Our results show that combining a stronger model with an explicit search structure yields performance gains. In other words, the stronger the model, the more effective the LFS framework is.
>
> While future long-context models may spontaneously adopt “self-corrective tokens” such as wait, however, or actually, this does not diminish the value of an explicit search scaffold in which the stronger models can operate.
>
> ### 2. “Handcrafted Exploration” and Where to Draw the Line
>
> We appreciate the reviewer’s nuanced question. When we state that LFS removes the need for “handcrafted exploration,” we are referring specifically to the necessity of manually choosing and tuning a search algorithm (BestFS vs MCTS vs. BFS vs. DFS) along with algorithm-specific hyperparameters such as the exploration constant in MCTS.
>
> In LFS:
>
> - we do not manually choose a search algorithm,
> - we do not manually tune exploration vs. exploitation,
> - and the LLM itself decides whether to continue, backtrack, or explore
>
> By contrast, MCTS, even when LLM-augmented, still requires a human-crafted search formula (UCB1), manual hyperparameter choices (e.g., c=0.5), and fixed update rules. These design decisions strongly influence its behaviour, as we saw in our results.
>
> Thus, LFS aims to strike a balance: providing structure while removing manual control. This “middle ground”, LLM-driven search within a lightweight scaffold, is precisely the contribution we aim to highlight.
>
> ### 3. Minor Weaknesses
>
> We thank the reviewer for these helpful corrections.
>
> - Line 652: “LLM-First- Search” → will address.
> - Line 222: “Three-of-Thoughts” → will address.
> - Page 27 formatting issue → will address.
>
> ### 4. Why does the MCTS tree appear wider?
>
> The reviewer’s intuition is correct: MCTS rollouts typically continue until a leaf node, resulting in possibly deeper trajectories. Our comment in Figure 31 referred specifically to the behaviour at the shown hyperparameter c. In this setting, MCTS repeatedly initiated exploration of new children rather than committing to promising branches. This led to a wider tree (many shallow exploratory expansions) rather than deeper rollouts.
>
> In other words, the width in the figure reflects the empirical behaviour at that particular hyperparameter setting, not a theoretical statement about MCTS depth.
>
> ### 5. Would MCTS improve with even smaller values of c?
>
> We saw that lower C values negatively effected CMTS in Countdown  but improved its ability in Sudoku. The two tasks differed substantially in sensitivity, and conducting a full sweep across tasks and models was not feasible given our compute budget, and there we chose what we thought was fair balance between the two.
>
> A key practical advantage of LFS is not needing to perform this type of hyperparameter search at all, the model determines its own exploration schedule.
>
> ### 6. Priority Queue Ties (Discrete LLM Scores)
>
> This is an good observation. Indeed, the LLM often assigns identical scores to multiple states because scores naturally cluster. This leads to ties in the priority queue.
>
> Our pragmatic solution was to choose the most recently inserted state among tied nodes (ordering preserved). This is a simple, deterministic tie-breaker that avoided additional complexity. The more “ideal” alternative, asking the LLM to re-rank all tied states, quickly became computationally intractable for large queues. We will clarify this behaviour further in the paper.

---

> > ### Comment · Reviewer_FYPb · 2025-11-24
> >
> > Thank you for your response; it addresses most of my concerns. Here are my follow-up questions:
> >
> > 1. Is the property of "commiting to promising branches" in question 4 of LFS coming from "choose the most recently inserted state among tied nodes (ordering preserved)" in question 6?
> >
> > 2. Is there any numerical results that the author could provide for their claim "we saw that lower C values negatively effected CMTS in Countdown but improved its ability in Sudoku"? I understand that a full sweep can be costly, but I would appreciate some empirical evidence.

---

> > > ### Author Response · Authors · 2025-11-25
> > >
> > > 1. The property of 'committing to promising branches' is not a result of the tie-breaking rule in the priority queue. Instead, it is an explicit architectural feature of the LFS framework.In LFS, the decision to 'commit' occurs before the priority queue is consulted. When the model deems the current state $s_t$ as promising, it bypasses the queue entirely and immediately generates the next step $s_{t+1}$ from s_t. The tie-breaking rule ('choose most recently inserted among tied nodes') only becomes active if the model decides the current branch is unpromising and chooses to query the priority queue for a new state.
> > >
> > > 2. As observed in Table 1, the optimal exploration constant varies by domain. In Countdown, c=1.0 yields slightly better performance than c=0.5 or c=1.5. Conversely, in Sudoku, c=0.5 significantly outperforms the higher values.We attribute this discrepancy to the structural differences between the two environments. Countdown is characterised by a wider, shallower search tree, which generally benefits from the higher exploration rate provided by c=1.0. Sudoku, however, requires a much deeper search where over-exploration is computationally costly. Consequently, we selected c=0.5 as the final parameter because it provides the necessary exploitation for Sudoku while maintaining competitive, albeit slightly suboptimal, performance in Countdown.

---

> > > > ### Comment · Reviewer_FYPb · 2025-11-25
> > > >
> > > > Thanks for the response; I have no other questions. I will keep my score for now and wait to see if the other reviewers' concerns are addressed.

---

### Official Review · Reviewer_KDbc · 2025-10-30

**Soundness:** 3
**Presentation:** 2
**Contribution:** 1
**Rating:** 2
**Confidence:** 3

**Summary:**

This paper introduces LLM-First Search (LFS), a novel method for reasoning and planning in Large Language Models (LLMs). The core premise of LFS is to replace the fixed search strategies and pre-defined hyperparameters (such as the exploration constant $C$ in MCTS) with an autonomous control mechanism managed directly by the LLM itself.

The LFS method employs the LLM in two key operations: (1) 'Evaluate', where the LLM assesses the value of all available actions from a given state, and (2) 'Explore', where the LLM dynamically decides whether to continue exploiting the current path or to "jump" to an alternative, high-potential path. To facilitate this, LFS maintains a priority queue of all unelected, promising actions, which serves as a mechanism for backtracking.

The authors evaluate LFS on two benchmarks, Countdown and Sudoku. The results reportedly show that LFS outperforms baseline methods, including ToT-BFS, BestFS, and MCTS, particularly on more difficult tasks, while also demonstrating superior computational efficiency.

**Strengths:**

1.  Well-Motivated Problem: The paper clearly identifies a significant limitation in existing search-augmented LLMs. It highlights the impracticality and sub-optimality of relying on fixed hyperparameters (like the MCTS exploration constant $C$), which require costly re-tuning for different tasks or models. The finding that MCTS performance can degrade when using a stronger model provides a compelling motivation for a more adaptive approach.

2.  Strong Empirical Performance: Within the confines of the two evaluated benchmarks, LFS demonstrates superior performance. The method achieves a higher WinRate than all baselines, with the performance gap widening on the most challenging tasks (e.G., Countdown Diff=7).

3.  Impressive Scalability and Efficiency: LFS shows strong evidence of scalability. It not only performs well but shows a greater performance improvement when paired with a stronger model (O3-mini) compared to baselines. Furthermore, the method is shown to be more computationally efficient, achieving a better Area Under Profile (AUP) for efficiency and generating smaller, more focused search trees than MCTS.

**Weaknesses:**

1.  Concerns Regarding Novelty: The proposed method appears to have significant overlap with the Tree-of-Thoughts (ToT) framework. The core mechanism—using an 'Evaluate' prompt for scoring and an 'Explore' prompt for decision-making, coupled with a priority queue—could be interpreted as a sophisticated form of prompt engineering built upon the ToT concept, rather than a fundamentally new search paradigm. The novelty beyond this implementation is not made sufficiently clear.


2.  Insufficient Analysis of Mechanism: The paper provides extensive data on *what* LFS achieves (i.e., higher WinRates) but lacks a deep analysis of *how* it achieves this performance. The claim that "self-guided" exploration works better is a high-level description of the phenomenon, not an explanation. There is no analysis of the LLM's decision-making process during the 'Explore' step. What cues does the LLM use to decide to backtrack? How do these emergent heuristics qualitatively differ from, and improve upon, the MCTS formula? Without this analysis, the paper is reporting numbers, not mechanisms.

3.  Limited Generalization: The validation is confined to two highly structured "toy examples" (Countdown and Sudoku), where states are clearly defined and rules are deterministic. No generalization is guaranteed beyond these benchmarks. The authors themselves acknowledge in the appendix that these tasks "lack some complexities of real-world problems". It is unclear if this "LLM-First" approach would be effective, or even feasible, in more open-ended, complex reasoning tasks where state representation is ambiguous.

**Questions:**

4.  Clarity and Presentation: The manuscript's structure is not always well-organized, making the central arguments difficult to follow at times. In particular, Figure 1 is hard to understand.

---

> ### Author Response · Authors · 2025-11-15
> **Response to Reviewer KDbc**
>
> ### 1. Concerns Regarding Novelty
>
> We thank the reviewer for raising this point. While LFS shares the high-level intuition that structured search can improve LLM reasoning, its core contribution is fundamentally different from Tree-of-Thoughts (ToT). ToT prescribes a fixed, externally defined search algorithm (BFS and DFS), with the LLM only supplying candidate thoughts. In contrast, LLM-First Search introduces a new search paradigm in which the model itself drives exploration, evaluation, and decision-making at every step. This shift, from structured prompting within a human-defined search algorithm to a method where the LLM itself is the search policy, is a meaningful conceptual contribution and not simply “sophisticated prompt engineering.”
>
> ### 2. Insufficient Analysis of Mechanism
>
> We appreciate the reviewer’s desire for deeper mechanistic insight. Our goal in this work was to establish the empirical effectiveness of self-guided exploration rather than to fully reverse-engineer the internal cues an LLM uses to navigate a search space. That said, our paper provides intuition about why LFS works:
>
> - the model uses natural-language evaluation to determine local promise,
> - and self-guided exploration and exploitation enables adaptive search depth and breadth
>
> Given the compute cost and the opacity of LLM internals, a full interpretability-style analysis of decision cues would require substantial additional development and budget, likely a standalone paper. We agree this is a fascinating direction for future work.
>
> However, we feel the paired combination of intuition + strong quantitative evidence is already sufficient to validate the proposed method, especially considering common practice in the literature for structured reasoning methods, which also emphasise empirical validation over low-level mechanistic analysis.
>
> ### 3. Limited Generalisation Beyond Two Tasks
>
> We appreciate the reviewer’s concern regarding external validity. While Countdown and Sudoku are indeed controlled environments, they have been used extensively in many influential papers (mentioned in section 5.2) to demonstrate algorithmic advances in LLM reasoning and planning. Moreover, these two benchmarks already represent distinct task families, arithmetic planning and constraint-based logic, which helps demonstrate that LFS is not tailored to a specific domain. The method itself is domain-agnostic, requiring only that one can (a) describe the state, (b) enumerate or generate candidate actions, and (c) interpret natural-language evaluations. There is nothing inherently task-specific in LFS that binds it to Countdown or Sudoku.

---

> > ### Comment · Reviewer_KDbc · 2025-11-26
> >
> > I would like to thank the authors for their response. I appreciate the clarification regarding the distinctions from ToT.
> >
> > However, I still maintain the view that the experimental evidence in the current manuscript is insufficient to fully validate the effectiveness of the proposed approach. To make the paper more convincing, I suggest the following additions:
> >
> > Qualitative Examples: Providing concrete qualitative examples that demonstrate exactly when and how the LLM-first search strategy is beneficial compared to baselines would be very persuasive.
> >
> > Analysis of Strengths and Weaknesses: A more granular analysis identifying specific situations and conditions where LFS exhibits its strengths versus its weaknesses is needed.
> >
> > I believe that incorporating these additional analyses would significantly strengthen the paper and provide the necessary depth to the empirical results.

---

> > > ### Author Response · Authors · 2025-12-01
> > >
> > > We thank the reviewer for the helpful suggestions. We have now incorporated both of the requested additions into the revised manuscript.
> > >
> > > First, we added a dedicated qualitative analysis section that provides concrete, step-by-step examples comparing LFS against MCTS, highlighting exactly where LFS's self-guided exploration offers a decisive advantage. These examples illustrate how LFS avoids common heuristic failures (such as local optima traps and premature abandonment of deep reasoning paths) and how its reasoning-driven exploration enables it to succeed in complex instances where MCTS fails.
> > >
> > > Second, we expanded the analysis to explicitly discuss the strengths and weaknesses of LFS across different task regimes. In particular, we clarify the conditions under which LFS and MCTS behave similarly (e.g., shallow or heuristically aligned tasks), and the settings where LFS demonstrates a clear structural advantage (e.g., wide, deep, or misleading search spaces where heuristics correlate poorly with solvability).
> > >
> > > We believe these additions address the reviewer’s concerns and significantly strengthen the empirical and conceptual clarity of the paper.

---

### Official Review · Reviewer_bV2f · 2025-11-01

**Soundness:** 1
**Presentation:** 2
**Contribution:** 1
**Rating:** 2
**Confidence:** 4

**Summary:**

The paper proposes LLM-First Search (LFS), a self-guided test-time search method where the language model itself decides when to exploit the current path or explore alternatives, scoring actions and dynamically backtracking via a priority queue and it removes hand-tuned exploration schedules and heuristics common in ToT-BFS, BestFS, and MCTS. Evaluated on Countdown and Sudoku with GPT-4o and o3-mini, LFS achieves competitive or superior WinRate, better efficiency (wins per token), and stronger scaling with harder tasks, larger models, and higher compute budgets.

**Strengths:**

1. Choosing win rate to accommodate generation stochasticity is sensible. The paper also reports Wilson 95% CIs, efficiency (wins per token), and performance profiles (AUP), which improves statistical transparency.

**Weaknesses:**

1. Countdown and Sudoku are rigorous but synthetic; standing alone (without math/coding or other reasoning tasks) they have relatively limited action spaces, which narrows external validity. The authors themselves note the restricted scope. However, given that it is a paper that proposes a methodology, the limitation is substantial.


2. Limited ablations: there is no analysis of LFS internals to understand the effectiveness of the proposed pipeline. Component-wise ablations would help isolate what drives gains.


3. Baselines seem incomplete: ToT-BFS baseline only appear in GPT-4o but not in o3-mini. MCTS for o3-mini only uses c=0.5 and as authors pointed out this particular choice was seen only optimal for CountDown. It might lead to unintentionally choosing weak baseline due the incompleteness of experiments. Furthermore, authors only evaluated on two models, which might be limited to validate the methodology.

**Questions:**

1. In Table 1/Figs., why is 6×6 Sudoku so low (e.g., with o3-mini: 0–25% WinRate across methods)? Do you suspect artifacts of the setup, and why not 9×9 to better reflect standard Sudoku difficulty?

---

> ### Author Response · Authors · 2025-11-15
> **Response to Reviewer bV2f**
>
> ### 1. Scope of Benchmarks (Countdown & Sudoku)
>
> We appreciate the reviewer’s concern regarding external validity. While Countdown and Sudoku are indeed controlled environments, they have been used extensively in many influential papers (mentioned in section 5.2) to demonstrate algorithmic advances in LLM reasoning and planning. Moreover, these two benchmarks already represent distinct task families, arithmetic planning and constraint-based logic, which helps demonstrate that LFS is not tailored to a specific domain. The method itself is domain-agnostic, requiring only that one can (a) describe the state, (b) enumerate or generate candidate actions, and (c) interpret natural-language evaluations. There is nothing inherently task-specific in LFS that binds it to Countdown or Sudoku.
>
> ### 2. Limited Ablations
>
> We agree that deeper component-wise ablations would shed insight on what aspects of LFS drive its performance. Due to academic compute constraints, we focused on demonstrating the overall effectiveness of the LFS pipeline across the selected tasks and models, rather than isolating each component. Full ablations would have quickly become computationally too expensive. That said, we acknowledge the value of such analyses and plan to include component-level ablations in future work, especially as more compute becomes available.
>
> ### 3. Baselines (ToT-BFS, MCTS variants, model coverage)
>
> Our compute budget imposed practical constraints on the number of baselines and hyperparameter sweeps we could run. As ToT consistently underperformed compared with other structured approaches in our preliminary experiments using GPT-4o, we elected not to further evaluate it on the stronger o3-mini model. Similarly, we used the MCTS hyperparameter c=0.5 because our initial results showed that for higher C values CountDown benefited but sudoku suffered and visa versa, and sweeping values for both tasks was computationally prohibitive. Our goal was to ensure baselines were representative rather than exhaustive, given the cost of running search-based methods on API models.
>
> ### 4. Why Performance on 6×6 Sudoku Is Low & Why Not 9×9
>
> 6×6 Sudoku appears low because the models struggled to complete the puzzle within the allowed token budget. In practice, the search often terminates before producing a complete, valid board. Given this difficulty, moving to 9×9 Sudoku would almost certainly yield even lower win rates (near zero), making it uninformative for comparative evaluation. Our choice was therefore driven by the desire to evaluate within a range where at least some methods have non-trivial solvability and where differences between approaches remain measurable.

---

> > ### Comment · Reviewer_bV2f · 2025-11-27
> >
> > Thank you for your detailed comments! I have no further questions,

---

### Author Response · Authors · 2025-12-01
**Meta-Response**

We thank the reviewers for their detailed and constructive feedback. We summarise here how all concerns have been addressed in the revised manuscript.

Reviewers highlighted several strengths of the work, including a well-motivated problem formulation, strong empirical performance on established reasoning benchmarks, clear evidence of scalability and efficiency, statistically transparent reporting, and comprehensive implementation details that improve reproducibility.

All major concerns have been resolved. Regarding external validity, we clarified that Countdown and Sudoku are standard and widely adopted benchmarks in the literature on structured reasoning with LLMs, and they represent distinct task families. This demonstrates that LFS is not tailored to a particular domain. We also emphasised that LFS is fully domain-agnostic, requiring only a state description, candidate actions, and natural-language evaluations.

Concerns regarding novelty were addressed by expanding the explanation of how LFS differs fundamentally from existing methods such as Tree of Thoughts. Unlike prior approaches that rely on fixed, externally defined search algorithms, LFS introduces a new search paradigm in which the LLM itself acts as the search policy and is responsible for exploration, exploitation, and backtracking. The priority queue serves only as a lightweight memory mechanism and not as a fixed traversal schedule.

To strengthen the analysis of how LFS achieves its performance, we added a dedicated qualitative analysis section that provides step-by-step comparisons between LFS and MCTS. We also introduced an explicit strengths-and-weaknesses analysis, describing task regimes where LFS aligns with traditional search behaviour and regimes where LFS offers clear advantages. These additions provide a deeper understanding of the method beyond quantitative results.

Regarding baselines, hyperparameter sweeps, and ablations, we clarified the compute-budget constraints inherent in API-based search methods and justified the decision to use representative settings rather than exhaustive sweeps. We also highlighted a key advantage of LFS, which is that it does not require any such per-task tuning.

All minor issues, including formatting, and typographical corrections have also been addressed.

Overall, the revised manuscript presents a conceptually distinct search paradigm, stronger empirical evidence, and additional qualitative and analytical insights requested during the discussion. We hope the AC will consider the concerns resolved and view the paper as meeting the threshold for acceptance.

---

### Meta-Review · Area_Chair_w7zo · 2025-12-16

**Summary:**

This paper proposes LLM-First Search (LFS), a method that delegates the "search policy itself" (such as switching between exploration/exploitation and backtracking) to the LLM, in contrast to reasoning approaches that utilize search techniques like Tree-of-Thoughts (ToT) or MCTS. The authors claim improvements in performance, efficiency, and scaling compared to ToT-BFS / BestFS / MCTS on the Countdown and Sudoku tasks.

However, two reviewers (bV2f, KDbc) expressed strong concerns regarding: (i) Weak external validity, as the evaluation is limited to only two tasks. (ii) Insufficient internal analysis regarding which elements are effective (lack of ablation studies or qualitative analysis). (iii) Insufficient explanation of novelty/differentiation (mainly KDbc) regarding the difference from existing methods like ToT. Consequently, both assigned a score of 2 (Reject).

The remaining reviewer (FYPb) acknowledged the paper's strengths and assigned a score of 6 (Marginally Above) but raised concerns and questions regarding positioning and design. After the author's response, FYPb explicitly stated they would "maintain the score for now." Overall, the prevailing judgment is that the current strength of evidence does not meet the acceptance threshold.

**Reviewer Concerns:**

* **External Validity (Limited Evaluation):** While Countdown and Sudoku are standard benchmarks, relying on only two tasks provides weak extrapolation to general tasks (**bV2f**, **KDbc**).
* **Lack of Internal Verification / "Why it works":** There is a lack of factor analysis within LFS, such as component ablation, qualitative examples of "Explore" decisions, and analysis of conditions under which the method is strong or weak (**bV2f**, **KDbc**). **KDbc** explicitly requested additional qualitative examples and a strength/weakness analysis.
* **Novelty & Positioning (Difference from ToT, etc.):** The framework of "Priority Queue + Evaluate/Explore" may appear as merely an implementation/prompt engineering extension of ToT, and the explanation of the conceptual difference is weak (mainly **KDbc**).
* **Validity of Baselines/Settings:** The lack of comprehensive sweeps, such as model coverage for ToT-BFS or search constant selection for MCTS, may affect the validity of the baseline strengths (**bV2f**).

**Reviewer Scores:**

* **Reviewer bV2f (Initial: 2):** There was no explicit statement of a score change in the discussion, and "None" was listed for additional questions. While the author's response provided some explanation regarding major concerns (external validity, ablation, baselines), strong backing such as additional experiments is not visible. Thus, prediction: **2 → 4 (High Probability)**.
* **Reviewer KDbc (Initial: 2):** While appreciating the explanation of the difference from ToT in the response, the reviewer stated that "addition of qualitative examples and strength/weakness analysis is necessary." Therefore, prediction: **2 → 2 (Medium-High Probability)**, or at most 2 → 4.
* **Reviewer FYPb (Initial: 6):** After the author's response, they explicitly stated they would "maintain the score for the time being while watching if other concerns are resolved." Thus, prediction: **6 → 6 Maintain (High Probability)**.

**Predicted Average:** (4 + 2 + 6) / 3 = **4** (Even in an optimistic case, roughly around **4.7**).

---

### Decision · Program_Chairs · 2026-01-26

Reject